# Temporal Difference Learning with Constrained Initial Representations

## Abstract

Recently, there have been numerous attempts to enhance the sample efficiency of off-policy reinforcement learning (RL) agents when interacting with the environment, including architecture improvements and new algorithms. Despite these advances, they overlook the potential of directly constraining the initial representations of the input data, which can intuitively alleviate the distribution shift issue and stabilize training. In this paper, we introduce the Tanh function into the initial layer to fulfill such a constraint. We theoretically unpack the convergence property of the temporal difference learning with the Tanh function under linear function approximation. Motivated by theoretical insights, we present our Constrained Initial Representations framework, tagged CIR, which is made up of three components: (i) the Tanh activation along with normalization methods to stabilize representations; (ii) the skip connection module to provide a linear pathway from the shallow layer to the deep layer; (iii) the convex Q-learning that allows a more flexible value estimate and mitigates potential conservatism. Empirical results show that CIR exhibits strong performance on numerous continuous control tasks, even being competitive or surpassing existing strong baseline methods.

## 1 Introduction

Conventional practice that aims at enhancing the sample efficiency of the reinforcement learning (RL) agent mainly focuses on algorithmic improvement, such as addressing the value overestimation issue (Fujimoto et al., 2018; Kuznetsov et al., 2020), improving the exploration (Haarnoja et al., 2018; Ladosz et al., 2022; Yang et al., 2024) or exploitation (Chen et al., 2021; D'Oro et al., 2023) capability of the agent, alleviating the primacy bias (Nikishin et al., 2022), leveraging model-based approaches (Buckman et al., 2018; Fujimoto et al., 2023; 2025) for planning (Hansen et al., 2022; 2024) or data augmentation (Voelcker et al., 2025), etc. Interestingly, these studies often rely on a small and simple network architecture (vanilla MLP in most cases). It appears that the RL community has long exhibited a preference for algorithmic refinement rather than architectural innovation. Recently, there have been some valuable attempts in scaling up RL algorithms and found that simply scaling up network capacity can degrade performance (Bjorck et al., 2021; Andrychowicz et al., 2020). BRO (Nauman et al., 2024) first successfully scales up network capacity and replay ratios by introducing strong regularization techniques, optimistic exploration, and distributional Q-learning (Bellemare et al., 2017). Furthermore, SimBa (Lee et al., 2025a) modifies the network architecture by injecting a simplicity bias and acquires strong performance across diverse domains.

Different from these advances, we provide a novel perspective to improve the sample efficiency, i.e., *directly constraining the input representations*. To that end, we incorporate the Tanh function into the initial layer of the network. This enjoys several advantages: (i) the Tanh function is increasing and order-preserving and does not change the sign of the representations; (ii) the Tanh function constrains the element to lie between $-1$ and $1$, ensuring value stabilization; (iii) it is easy to compute its gradient. The Tanh function can also help alleviate the severe distribution shift caused by OOD input samples, as illustrated in Figure 1. To better support our claim, we consider a two-layer MLP network $y = W_2(\sigma(W_1 x))$, where we omit the bias term for simplicity, $x \in \mathbb{R}^d$ is the input vector, $W_1 \in \mathbb{R}^{h \times d}, W_2 \in \mathbb{R}^{1 \times h}$ are weight matrices, $\sigma(\cdot)$ denotes the activation function, $h$ is the hidden dimension, $d$ is the input dimension. If $\sigma(\cdot) = \text{ReLU}(\cdot)$, then given the input $x_1$ and $x_2$, the output deviation gives $|y_1 - y_2| \leq \|W_2\|_F \cdot \|W_1\|_F \cdot \|x_1 - x_2\|_F$, where $\|\cdot\|_F$ is the Frobenius norm. Unfortunately, the above upper bound is *unconstrained* and can be large if $x_2$ deviates far from $x_1$

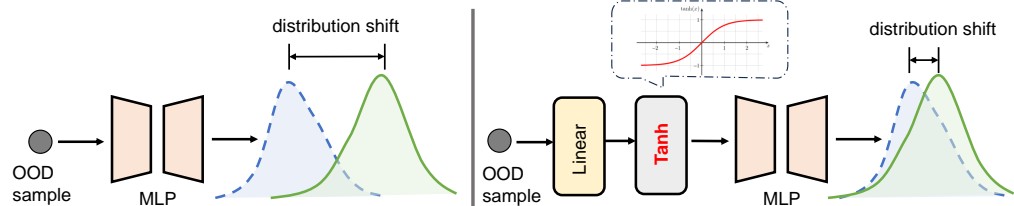

Figure 1: **Illustration of our motivation.** When encountering the out-of-distribution (OOD) sample that deviates *far* from the distribution of the current policy, the vanilla MLP network may incur severe distribution shift issue (**left**) while after adding the `Tanh` activation to the MLP, the negative influence of the OOD sample can be mitigated (**right**).

(this can often happen due to exploration and evolving policy). If $\sigma(\cdot) = \tanh(\cdot)$, we similarly have $|y_1 - y_2| \leq \|W_2\|_F \cdot \|\tanh(W_1 x_1) - \tanh(W_1 x_2)\|_F$. Since $|\tanh(x)| \leq 1$, the upper bound here can always be bounded regardless of how $x_2$ differs from $x_1$ (the output will not change drastically). That being said, the negative influence of the potential OOD samples can be mitigated.

To further show the benefits of using the `Tanh` function, we conduct theoretical analysis on temporal difference (TD) learning with the `Tanh` function under linear function approximation. Our results show that the linear independence between basis functions can still hold after using the `Tanh` function. Another exciting finding is that incorporating the `Tanh` function can reduce the variance of the gradient of the least square TD (LSTD) objective function, which validates that the `Tanh` function can stabilize training. Furthermore, we show that TD(0) can still converge to a fixed point after adding the `Tanh` function and the global convergence can be guaranteed by adding regularizations. These theoretical results pave the way for applying the proposed idea to the DRL setting.

As a result, we propose the Constrained Initial Representations algorithm, namely CIR. It mainly contains three key components: (i) the representation constraint module that leverages the `Tanh` function, the average representation normalization and layer normalization (Ba et al., 2016) to stabilize representations and mitigate the possible gradient vanishing issue; (ii) the skip connection module that maintains a direct information pathway from the shallow layer to the deep layer to better pass information; (iii) the convex Q-learning that permits better value estimate and mitigates the risk of conservatism from `Tanh`. These components are novel and feature distinct ways of achieving high sample efficiency compared to previous methods. We evaluate CIR on DMC suite (Tassa et al., 2018), HumanoidBench (Sferrazza et al., 2024) and ODRL (Lyu et al., 2024c) tasks. The experimental results show that CIR can match or even outperform recent strong baseline methods on numerous tasks. To facilitate reproducibility, we include our codes in the supplemental materials.

## 2 RELATED WORK

**Regularization in DRL.** Introducing implicit or explicit regularization terms has been well-explored in other fields (Tang et al., 2018; Xu et al., 2015; Xie et al., 2016; Xin et al., 2021; Merity et al., 2017). In DRL, there are also many studies that add new regularization terms into the critic or the actor network to stabilize training, including regularizing the uncertainty of value estimate (Lyu et al., 2022; Eysenbach et al., 2023), penalizing the TD error of the critic (Parisi et al., 2018; Shao et al., 2022), involving action gradient regularization (Kostrikov et al., 2021; D'Oro & Jaśkowski, 2020; Li et al., 2022), divergence regularization (Su & Lu, 2021), mutual information regularization (Leibfried & Grau-Moya, 2019), etc. Besides, regularization techniques from supervised learning are also widely adopted in DRL to alleviate the overfitting issue, e.g., normalization tricks (Ba et al., 2016; Gogianu et al., 2021; Bjorck et al., 2021; Bhatt et al., 2024; Lee et al., 2025a;b; Palenicek et al., 2025; Elsayed et al., 2024), dropout (Srivastava et al., 2014; Wu et al., 2021; Hiraoka et al., 2021), weight decay (Farebrother et al., 2018; Nauman et al., 2024), etc. Moreover, some researchers resort to reinitialization approaches to escape from the local optima and boost the sample efficiency of the agent, e.g., resetting the parameters to their initial distributions (Nikishin et al., 2022; Qiao et al., 2024), selectively reinitializing dormant weights (Sokar et al., 2023), combining current weights with random weights (Schwarzer et al., 2023; Xu et al., 2024). In contrast, our CIR framework directly regularizes the *initial representations* of the transition sample.

**Sample-efficient RL algorithms.** A long-standing goal in DRL is to enable the agent to quickly acquire strong continuous control policies with a small budget of data from the environment (Yu, 2018; Du et al., 2019). To fulfill that, some researchers enhance the sample efficiency by improving the exploration ability of the agent (Asmuth et al., 2012; Still & Precup, 2012; Burda et al., 2018; Haarnoja et al., 2018; Ladosz et al., 2022; Yang et al., 2024; Jiang et al., 2025), reducing value estimation bias (Van Hasselt et al., 2016; Fujimoto et al., 2018; Kuznetsov et al., 2020; Moskovitz et al., 2021; Lyu et al., 2022; 2023), using new experience replay methods (Fujimoto et al., 2020; Schaul et al., 2015; Andrychowicz et al., 2017), increasing the frequency of data reuse (Chen et al., 2021; D'Oro et al., 2023; Lyu et al., 2024b; Romeo et al., 2025), injecting representation learning objectives (De Bruin et al., 2018; Ota et al., 2020; Laskin et al., 2020; Fujimoto et al., 2023; Yan et al., 2024; Fujimoto et al., 2025), leveraging model-based approaches (Janner et al., 2019; Buckman et al., 2018; Hansen et al., 2022; Hafner et al., 2023; Hansen et al., 2024; Voelcker et al., 2025), etc. The above research can be categorized into algorithmic improvement. Another line of studies focuses on architecture improvement, which has been widely studied in computer vision (CV) (He et al., 2015; Targ et al., 2016; Goodfellow et al., 2021; Ho et al., 2020) and natural language processing (NLP) (Vaswani et al., 2017; Devlin et al., 2019; Beck et al., 2024). BRO (Nauman et al., 2024) empirically shows that architecture modification can incur strong performance improvement, breaking the curse that simply scaling network capacity can degrade performance (Andrychowicz et al., 2020; Bjorck et al., 2021). Moreover, the SimBa architecture (Lee et al., 2025a) outperforms BRO by incorporating simplicity bias into the network design. Compared to these studies, our CIR method also modifies the network architecture with a `Tanh` function to stabilize representations.

## 3 PRELIMINARIES

Reinforcement learning (RL) problems can be formulated by the tuple $(\mathcal{S}, \mathcal{A}, r, \gamma, P)$, where $\mathcal{S}$ is the state space, $\mathcal{A}$ is the action space, $r(s, a) : \mathcal{S} \times \mathcal{A} \to \mathbb{R}$ is the scalar reward signal, $\gamma \in [0, 1]$ is the discount factor, $P(s'|s, a)$ is the transition probability. In online RL, the agent continually interacts with the environment using a policy $\pi_\psi(\cdot|s)$ parameterized by $\psi$. The goal of the agent is to maximize the expected cumulative discounted rewards: $\max J(\psi) = \mathbb{E}[\sum_{t=0}^{\infty} \gamma^t r(s_t, a_t)|s_0, a_0; \pi]$. We have the state-action value function $Q^\pi(s, a) := \mathbb{E}_\pi[\sum_{t=0}^{\infty} \gamma^t r(s_t, a_t)|s_0 = s, a_0 = a]$, the value function $V^\pi(s) := \mathbb{E}_\pi[\sum_{t=0}^{\infty} \gamma^t r(s_t, a_t)|s_0 = s]$.

For theoretical analysis, we assume that the MDP has finite state space and action space with size $S$ and $A$, and the Markov chain in this MDP has a stationary distribution $\nu$ and is ergodic. We consider linear function approximation (Tsitsiklis & Van Roy, 1996; Bhandari et al., 2018; Jin et al., 2020; Zou et al., 2019), where the value function $V^\pi$ is a linear combination of features, $V(s; \theta) = \phi(s)^\mathsf{T}\theta$, where $\phi : \mathcal{S} \to \mathbb{R}^d$ is a known feature map, $\theta$ is the weight vector. We similarly have the state-action value function $Q(s, a; \theta) = \psi(s, a)^\mathsf{T}\theta$, $\psi : \mathcal{S} \times \mathcal{A} \to \mathbb{R}^d$ is the feature map. We focus on $V$ for convenience and consider the TD(0) update rule, which is given by

$$\theta_{t+1} \leftarrow \theta_t + \alpha_t(r_t + \gamma\phi(s_{t+1})^\mathsf{T}\theta_t - \phi(s_t)^\mathsf{T}\theta_t)\phi(s_t), \tag{1}$$

where $\alpha_t$ is the step size at timestep $t$. We then introduce per-element `Tanh` function to the feature and have $V(s; \theta) = \tanh(W\phi(s))^\mathsf{T}\theta$, where $W \in \mathbb{R}^{d \times d}$ is a positive definite diagonal constant matrix. Similarly, we have the TD(0) update rule after introducing $\tanh$,

$$\theta_{t+1} \leftarrow \theta_t + \alpha_t(r_t + \gamma\tanh(W\phi(s_{t+1}))^\mathsf{T}\theta_t - \tanh(W\phi(s_t))^\mathsf{T}\theta_t)\tanh(W\phi(s_t)). \tag{2}$$

Without loss of generality, we assume that the rewards are bounded, $\forall s, a, |r(s, a)| \leq r_{\max}$. Given a positive definite matrix $D$, we denote $\|x\|_D = \sqrt{x^\mathsf{T}Dx}$ as the vector norm induced by $D$. For simplicity, we denote $\|\cdot\| = \|\cdot\|_I$, where $I$ is the identity matrix.

## 4 THEORETICAL ANALYSIS

In this section, we theoretically analyze the benefits of incorporating the `Tanh` function for representation stabilization and convergence properties in temporal difference learning. We conduct theoretical analysis under linear function approximation since it can be difficult to analyze its theoretical properties when involving neural networks. Due to space limit, all proofs are deferred to Appendix A. We first impose the following assumption, which is widely adopted in previous literature (Tsitsiklis & Van Roy, 1996; Jin et al., 2020; Zhang et al., 2023).

**Assumption 4.1.** *The feature matrix $\phi = [\phi_1, \phi_2, \ldots, \phi_S]^\mathsf{T}$ has full column rank, i.e., the basis functions $\{\phi_k | k = 1, \ldots, S\}$ are linearly independent.*

The above assumption is critical for convergence analysis. We show in Theorem 4.2 that this property still holds for the feature matrix after applying the `Tanh` function by properly choosing $W$.

**Theorem 4.2** (Linear Independence). *Suppose that $W = c \cdot I$, where $c \in \mathbb{R}$ is a sufficient small positive number, $I$ is the identity matrix, then if basis functions $\{\phi_1, \ldots, \phi_S\}$ are linearly independent, then the transformed basis functions $\{\tanh(W\phi_1), \ldots, \tanh(W\phi_S)\}$ are also linearly independent.*

Furthermore, we show in the following theorem that the constrained features can incur a smaller gradient variance compared to its unconstrained counterpart.

**Theorem 4.3** (Variance Reduction). *Given any $s$, suppose the feature $\phi(s)$ follows a distribution with $\mathbb{E}[\phi(s)] = 0$. Consider per-transition least square TD (LSTD) objective $\mathcal{L} = \frac{1}{2}[(r + \gamma V(s'; \theta) - V(s; \theta))^2]$, then if $V(s; \theta) = \phi(s)^\mathsf{T}\theta$, the variance of the semi-gradient term satisfies:*

$$\|\mathrm{Var}(\nabla_\theta \mathcal{L})\| \leq (r_{\max} + (1 + \gamma)\|\theta\|\Lambda)^2 \cdot \|\phi(s)\|^2, \tag{3}$$

*where $\Lambda = \max\{\|\phi(s)\|, \|\phi(s')\|\}$. While if $V(s; \theta) = \tanh(W\phi(s))^\mathsf{T}\theta$, we have*

$$\|\mathrm{Var}(\nabla_\theta \mathcal{L})\| \leq (r_{\max} + (1 + \gamma)\|\theta\|\lambda_W\Lambda)^2 \cdot \|\tanh(W\phi(s))\|^2, \tag{4}$$

*where $\lambda_W$ is the maximum eigenvalue of $W$.*

**Remark.** By properly choosing the constant matrix $W$, the upper bound of the variance term with the `Tanh` function can be smaller than the upper bound of the vanilla variance term. For example, by setting $W = I$ (identity matrix), $\tanh(W\phi(s)) = \tanh(\phi(s))$, $\lambda_W = 1$, then we have

$$(r_{\max} + (1 + \gamma)\|\theta\|\lambda_W\Lambda)^2 \cdot \|\tanh(\phi(s))\|^2 \leq (r_{\max} + (1 + \gamma)\|\theta\|\Lambda)^2 \cdot \|\phi(s)\|^2,$$

where we use the fact that $|x| \geq |\tanh(x)|, \forall x$. That is, if one chooses $W = c \cdot I, c \in \mathbb{R}, c \in (0, 1)$, the upper bound after using `Tanh` is always tighter. It indicates that `Tanh` *can stabilize the training process and incur smaller gradient variance*. Note that we do not make any specific assumptions on $\|\phi\|$ since the `Tanh` function enforces an inherent constraint on the features. Since $|\tanh(x)| \leq 1, \forall x$, we have $\|\tanh(W\phi(s))\| \leq \sqrt{d}$, then the variance upper bound can be simplified,

$$\|\mathrm{Var}(\nabla_\theta \mathcal{L})\| \leq (r_{\max} + (1 + \gamma)\|\theta\|\sqrt{d})^2 \cdot d. \tag{5}$$

We then formally show that TD(0) with linear function approximation and `Tanh` feature transformation is guaranteed to converge to a fixed point.

**Theorem 4.4** (Convergence under TD(0)). *Consider the TD(0) update rule for the value function $V(s; \theta) = \tanh(W\phi(s))^\mathsf{T}\theta$. Suppose we project the parameters $\theta$ onto bounded sets $\Theta = \{\theta \mid \|\theta\|_2 \leq C_\theta\}$ after each update. If the step sizes satisfy $\sum \alpha_t = \infty$, $\sum \alpha_t^2 < \infty$, then the projected TD(0) algorithm converges almost surely to a fixed point $\theta^*$.*

However, the above theorem can only ensure that the weight converges to a fixed point. It remains unclear whether it can converge to a global optimum. We then introduce the regularized optimization objective and show that $\theta$ converges to a unique global minima $\theta^*$.

**Theorem 4.5** (Global Convergence under Regularization). *Given the regularized loss $\mathcal{L}(\theta) = \mathbb{E}\left[\left(r + \gamma\tanh(W\phi(s'))^\mathsf{T}\theta - \tanh(W\phi(s))^\mathsf{T}\theta\right)^2\right] + \lambda_\theta\|\theta\|_2^2$, assume that (a) the step size $\alpha$ is a constant; (b) the gradient of the loss $\mathcal{L}(\theta)$ is $\beta$-smooth, i.e., $\|\nabla\mathcal{L}(\theta_1) - \nabla\mathcal{L}(\theta_2)\| \leq \beta\|\theta_1 - \theta_2\|$; (c) the regularization parameter satisfies $0 < \lambda_\theta < \frac{\beta}{2}$. Then, gradient descent with step size $\alpha \leq 1/\beta$ converges linearly to a unique global minimum $\theta^*$.*

**In summary**, the above theoretical results generally describe the variance reduction capability and the convergence results after applying the `Tanh` function on features, which sheds light on extending this simple idea into the DRL scenario.

## 5 ALGORITHM

Motivated by the theoretical results, we formally present our CIR framework, which mainly comprises three parts: (i) a `Tanh` function and two normalization techniques to stabilize representations; (ii) the skip connection module that combines information from shallow layers and deep layers; (iii) a convex Q-learning approach to further enhance the performance of the agent.

## 5.1 CONSTRAINING INITIAL REPRESENTATION WITH THE TANH ACTIVATION

Our focus is on state-based continuous control tasks, where plain MLP networks are commonly used. However, it can be fragile to samples that deviate far from the current distribution (as analyzed in Section 1). To stabilize training, we add a Tanh function to the initial layer of the network. We do not place the Tanh function in the middle layer or deep layers because (i) the latter layers can be naturally regularized if the initial representations are constrained, since the magnitude of the input representations are controlled, and (ii) it may be challenging for the gradient to backpropagate and the gradient may vanish before the initial layers can be well-updated.

Nevertheless, naively utilizing the Tanh function can incur gradient vanishing issue (Wang et al., 2019; Goodfellow et al., 2016), e.g., given the input vector $\mathbf{o}_t \in \mathbb{R}^d$ at timestep $t$, if the initial representations $\mathbf{x}_t = \text{Linear}(\mathbf{o}_t) \in \mathbb{R}^h$ have large values (i.e., $\tanh(\mathbf{x}_t) \approx [1, 1, \ldots, 1]$), its gradient after applying the Tanh function can be small, $\nabla \tanh(\mathbf{x_t}) \approx \mathbf{0}$ (since $\nabla \tanh(x) = 1 - (\tanh(x))^2$). To mitigate this concern, we need to manually compress the representations to ensure that they do not lie in the gradient vanishing region of Tanh. A natural idea is to weight the initial representations with a small constant $c$ (this can be seen as setting the weight matrix $W = c \cdot I$ in our theoretical analysis). However, this approach lacks flexibility because different environments possess different state space and action space dimensions, dynamics transition probability, and reward scales. One may need to grid search the best $c$ per task, which can be labor-intensive and expensive. Instead, we introduce the average representation normalization trick (AvgRNorm) to adaptively control the representations from different tasks without tuning the parameter $c$,

$$\text{AvgRNorm}(\mathbf{x}) = c \times \frac{\mathbf{x}}{\text{Mean}(\tilde{\mathbf{x}})}, \quad \tilde{\mathbf{x}} = |\mathbf{x}|, \tag{6}$$

where $\text{Mean}(\cdot)$ is the mean operator, $c$ is a small positive number that controls the scale of the normalized representations (as required in Theorem 4.2). We use the absolute values of representations to avoid $\text{Mean}(\tilde{\mathbf{x}}) \approx 0$. AvgRNorm can effectively bound representations and decrease the gradient contribution of those large values in representations. We adopt $c = 0.1$ by default.

Furthermore, based on Theorem 4.5, additional regularization terms are needed to acquire global convergence. Since regularizing the parameters is inherently equivalent to regularizing the representations, we further regularize representations for better stability by involving layer normalization, which is proven effective in prior works (Lee et al., 2025a; Nauman et al., 2024). Finally, the constrained initial representation is given by,

$$\mathbf{z}_t = \tanh(\text{AvgRNorm}(\text{LayerNorm}(\text{Linear}(\mathbf{o}_t)))). \tag{7}$$

## 5.2 BOOSTING GRADIENT FLOW WITH SKIP CONNECTIONS

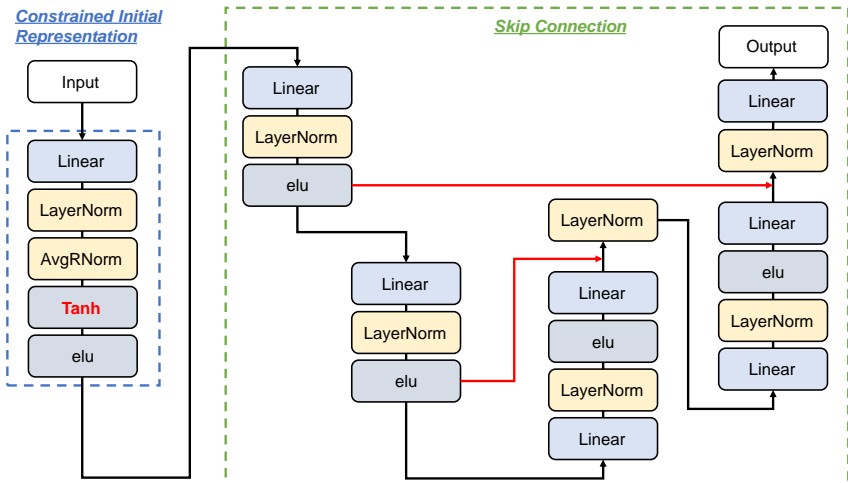

Figure 2: **Architecture overview of CIR.** CIR adopts (a) the AvgRNorm module and the Tanh activation to stabilize initial representations; (b) skip connection modules to facilitate gradient flow.

After the initial representations are constrained, we feed forward them to downstream layers. Built upon previous findings that scaling aids sample efficiency (Nauman et al., 2024; Lee et al., 2025a;b), we also scale up the critic networks by expanding the depth and width of the critic networks. To better facilitate gradient propagation, we utilize the skip connection technique (Ronneberger et al., 2015; Milletari et al., 2016), which is less explored in DRL. That indicates that we adopt a U-shape network, which contains a down-sampling path to capture features, and a symmetric up-sampling path to aggregate features from shallow layers and deep layers. Different from prior works, we do not expand the channel of the features during the process of either down-sampling or up-sampling, and we fuse shallow and deep abstract representations through linear addition rather than concatenation.

We still adopt layer normalization for stabilizing representations in the latter layers (which can be seen as the regularization term in Theorem 4.5). Formally, for the down-sampling layers $l \in \{1, \ldots, L\}$, with the number of layers $L$, its representations at timestep $t$ is given by,

$$\mathbf{x}_t^l = \text{LayerNorm}(\text{Linear}(\mathbf{x}_t^{l-1})), \tag{8}$$

where $\mathbf{x}_t^0 = \mathbf{z}_t$ is the initial constrained representation calculated via Equation 7. For a symmetric up-sampling layer $l \in \{1, \ldots, L\}$, its representation is calculated by linear addition, i.e.,

$$\mathbf{x}_t^{2L-l+1} = \mathbf{x}_t^l + \text{Linear}(\text{LayerNorm}(\text{Linear}(\mathbf{x}_t^{2L-l}))). \tag{9}$$

This resembles residual learning (He et al., 2015). Note that we omit the activation function terms in Equation 8 and Equation 9 for clarity (we use the `elu` activation function). There are two reasons for adopting the skip connection technique. First, the U-shape network with skip connections enables effective multi-scale feature extraction, facilitating both global feature understanding and local information preservation. Second, it combines shallow low-level information with deep abstract representations, which enables model scaling and eases gradient vanishing. It also enhances the representational capacity of the model. Meanwhile, we do not involve modules like convolution layers or pooling. The detailed network architecture is shown in Figure 2. We use 2 down-sampling and up-sampling layers by default ($\approx$ 3M parameters). We adopt the vanilla MLP network for the actor since we find that modifying network architecture or scaling the actor network is less effective, which matches the results in BRO (Nauman et al., 2024).

## 5.3 CONVEX Q-LEARNING

Following our theoretical results, we adopt the TD(0)-style update rule for training critic networks. Denote $Q_{\theta_1}(s, a), Q_{\theta_2}(s, a)$ as the critic networks parameterized by $\theta_1, \theta_2$, respectively, $\pi_\phi(\cdot|s)$ as the actor network parameterized by $\phi$, the objective function of TD(0) with clipped double Q-learning (CDQ) gives $\mathcal{L}(\theta_i) = \mathbb{E}_{(s,a,r,s') \sim \rho}[(y - Q_{\theta_i}(s, a))^2]$, where $y = r + \gamma(\min_{j \in \{1,2\}} Q_{\theta'_j}(s', a') - \alpha \log \pi_\phi(a'|s'))$ is the target value, $\theta'_j, j \in \{1, 2\}$ are target network parameters, $a' \sim \pi_\phi(\cdot|s)$, $\rho$ is the sample distribution in the replay buffer, $\alpha$ is the temperature coefficient. Existing studies show that CDQ can incur underestimation bias (i.e., conservatism) (Ciosek et al., 2019; Pan et al., 2020; Lyu et al., 2022). Meanwhile, constraining initial representations with `Tanh` can be conservative since it constrains the initial representations and the output of the model. Eventually, the conservatism may be excessive if we use CDQ in CIR.

To mitigate the potential negative influence of conservatism from the architecture, we propose a convex Q-learning approach to encourage the learned value function to be optimistic. In convex Q-learning, we modify the target value by combining the minimum and maximum of value functions,

$$y = r(s, a) + \gamma \left( \lambda \times \min_{i \in \{1,2\}} Q_{\theta'_i}(s', a') + (1 - \lambda) \times \max_{i \in \{1,2\}} Q_{\theta'_i}(s', a') - \alpha \log \pi_\phi(a'|s') \right). \tag{10}$$

The over-conservatism issue can be eased by properly choosing $\lambda$. We set $\lambda = 0.3$ by default. We show below that the convex Q-learning is guaranteed to converge to the optimal value function.

**Theorem 5.1.** *Under mild assumptions, for any $\lambda \in (0, 1]$, the Convex Q-learning converges to the optimal value function $Q^*$ as defined by the Bellman optimality equation with probability 1.*

We use average $Q$ values for updating the actor following prior works (Chen et al., 2021; Nauman et al., 2024), i.e., $\mathcal{L}(\phi) = \mathbb{E}_{s \sim \rho, \tilde{a} \sim \pi_\phi(\cdot|s)}[\alpha \log \pi_\phi(\tilde{a}|s) - \frac{1}{2} \sum_{j=1,2} Q_{\theta_j}(s, \tilde{a})]$. We further adopt the sample multiple reuse (SMR) (Lyu et al., 2024b) trick that reuses the fixed sampled batch $M$ times to boost sample efficiency, which functions better than other data replay methods.

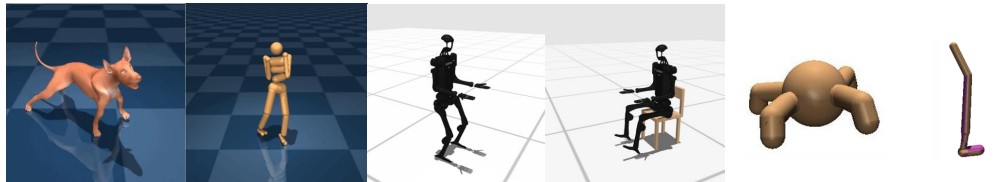

Figure 3: **Visualizations of the benchmarks.** We consider tasks from the DMC suite, Humanoid-Bench and ODRL for evaluations. These tasks feature varying complexity and can be challenging.

Putting together all the aforementioned techniques gives birth to our CIR algorithm. We adopt SAC (Haarnoja et al., 2018) as the base algorithm, modifying its critic network architecture and replacing CDQ with the convex Q-learning. The pseudo-code for CIR is available in Appendix B.

# 6 EXPERIMENTS

In this section, we empirically investigate the applicability of CIR across numerous tasks. Across each domain, we evaluate the performance of CIR under a fixed set of hyperparameters and algorithmic configurations[1]. Due to space limits, more empirical results are deferred to Appendix E.

We utilize three continuous control benchmarks for performance evaluation, DMC suite (Tassa et al., 2018), HumanoidBench (Sferrazza et al., 2024) and ODRL (Lyu et al., 2024c), as depicted in Figure 3. We adopt 20 DMC suite easy and medium-level tasks, 7 DMC hard tasks, 14 HumanoidBench locomotion tasks, and 8 ODRL tasks, leading to a total of 49 tasks. Following SimBa (Lee et al., 2025a), we use different training steps per environment based on their complexity, 500K steps for DMC easy and medium tasks, 1M for DMC hard and ODRL tasks, 2M for HumanoidBench tasks. Details on environmental descriptions can be found in Appendix C. Note that in ODRL, there exist dynamics discrepancies between the source domain and the target domain, and the agent needs to acquire good performance in the target domain by using data from both domains. This benchmark is adopted to empirically evaluate CIR's ability to handle OOD data, as illustrated in Figure 1.

## 6.1 MAIN RESULTS

We consider the following methods as baselines: **SAC** (Haarnoja et al., 2018), a classical and widely used off-policy maximum entropy RL algorithm; **TD7** (Fujimoto et al., 2023), an enhanced version of TD3 (Fujimoto et al., 2018) by incorporating state-action representation learning; **BRO** (Nauman et al., 2024) that combines critic network scaling with layer normalization and residual layer, optimistic exploration, distributional value modeling, and periodic resets; **DreamerV3** (Hafner et al., 2023), a strong model-based RL algorithm that optimizes a policy via synthetic trajectories; **TD-MPC2** (Hansen et al., 2024) that empowers policy learning by training latent world models and leveraging model predictive control; **SimBa** (Lee et al., 2025a) that directly modifies the network architecture of the agent by combining running statistics normalization (RSNorm), a residual feed-forward block, and post-layer normalization. For a fair comparison, we use BRO-Fast and SimBa with the update-to-data (UTD) ratio 2. We use the SMR ratio $M = 2$ in CIR across all tasks.

For ODRL tasks, we consider the following baselines: **SAC** (Haarnoja et al., 2018), **DARC** (Eysenbach et al., 2021) that train domain classifiers for cross-dynamics policy adaptation, and **PAR** (Lyu et al., 2024a) that fulfill efficient policy adaptation via capturing representation mismatch.

We summarize empirical results in Figure 4, where the x-axis denotes the computation time using an RTX 3090 GPU, and the y-axis represents the aggregated return. Points that approach the upper-left region indicate higher compute efficiency while points in the lower-right region have lower compute efficiency. The shaded region denotes standard deviations. We find that CIR exhibits strong performance on DMC easy and medium tasks, exceeding all baselines. On HumanoidBench tasks, CIR outperforms strong baselines like SimBa, and matches the performance of TD-MPC2. For DMC hard tasks, CIR underperforms SimBa and BRO, but is still better than other baselines. It is evident that CIR can achieve competitive sample efficiency against recent strong model-free methods while

---

[1]Methods like SimBa (Lee et al., 2025a) adopt different algorithmic configurations on different tasks

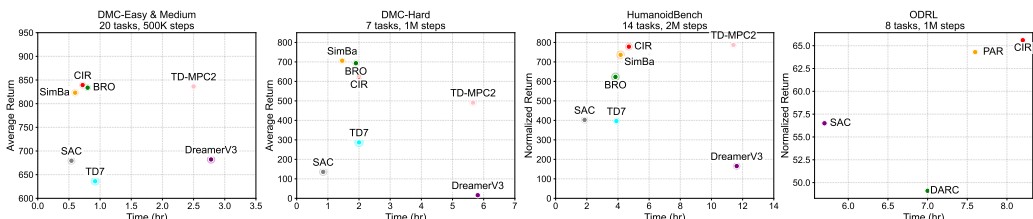

Figure 4: **Comparison of CIR against baselines.** The average episode return is compared on DMC tasks while normalized return results are compared in HumanoidBench and ODRL tasks.

not increasing much computation time. CIR is 2 times computationally more efficient compared to model-based methods like TD-MPC2 and DreamerV3. Moreover, CIR exhibits strong performance on ODRL tasks, despite that CIR does not include any special components designed for this setting, indicating its advantages in mitigating the OOD issue and stabilizing training. The empirical results highlight the effectiveness of CIR and the potential of constraining initial representations.

## 6.2 EXTENDED RESULTS ON COMPLEX TASKS

Though CIR underperforms SimBa on some tasks, it can beat SimBa on complex tasks like HumanoidBench. To further show the effectiveness of CIR, we conduct experiments on 2 HumanoidBench loco-motion tasks without dexterous hands and 4 locomotion tasks with dexterous hands. Episode return results in Table 1 show that CIR consistently beats SimBa, sometimes by a significant margin. This further verifies the advantages of using CIR.

Table 1: **Comparison of CIR and SimBa on wider HumanoidBench tasks.** We bold the best score.

| Task | SimBa | CIR |
|---|---|---|
| h1-door | 294.3±21.3 | **327.2**±11.3 (**+11.2%**) |
| h1-highbar_simple | 491.0±27.2 | **512.8**±29.6 (**+4.4%**) |
| h1hand-reach | 5581.7±789.9 | **5776.6**±357.3 (**+3.5%**) |
| h1hand-run | 31.0±5.0 | **193.7**±108.6 (**+524.8%**) |
| h1hand-sit_simple | 755.5±104.0 | **856.9**±65.2 (**+13.4%**) |
| h1hand-sit_hard | 559.4±205.6 | **678.5**±160.2 (**+21.3%**) |

## 6.3 ABLATION STUDY

In this part, we conduct a detailed ablation study of CIR. We first adopt 7 DMC medium tasks and 7 DMC hard tasks for experiments, and consider (i) **NoTanh**, where we discard the `Tanh` function in CIR; (ii) **NoLN**, a variant of CIR without layer normalization; (iii) **CDQ**, which utilizes CDQ instead of the convex Q-learning; (iv) **UTD**, which replaces SMR with UTD in CIR (UTD = 2). We summarize the results in Figure 5 (left), where we report the percent CIR performance (% CIR performance) of these variants. If the percent CIR performance exceeds 100, it indicates that the variant outperforms CIR, and underperforms CIR if it is smaller than 100. We further use 20 DMC easy & medium tasks for experiments and consider (v) **Sigmoid**, (vi) **Softmax**, (vii) **LayerNorm**, that replace `Tanh` with sigmoid, softmax, and layer normalization, respectively; (viii) **NoSC**, which removes the skip connection module in CIR. The results are presented in Figure 5 (right).

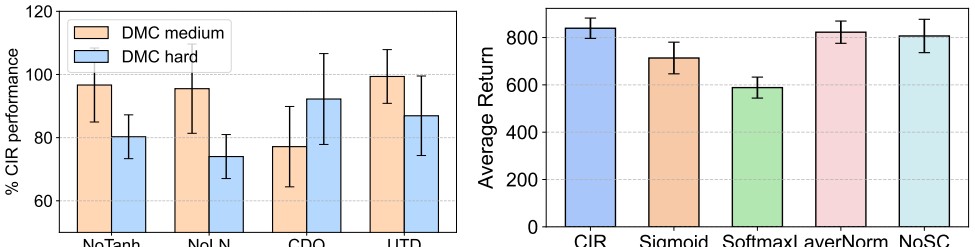

Figure 5: **Left:** Ablation study on network components and algorithmic components in CIR. **Right:** Ablation study on `Tanh` and skip connection module in CIR. SC denotes skip connection.

**On the critic network architecture.** It turns out that eliminating either the `Tanh` function or layer normalization decreases the performance of CIR, especially on complex DMC hard tasks. The `Tanh` function is responsible for initial representation constraint, while layer normalization ensures

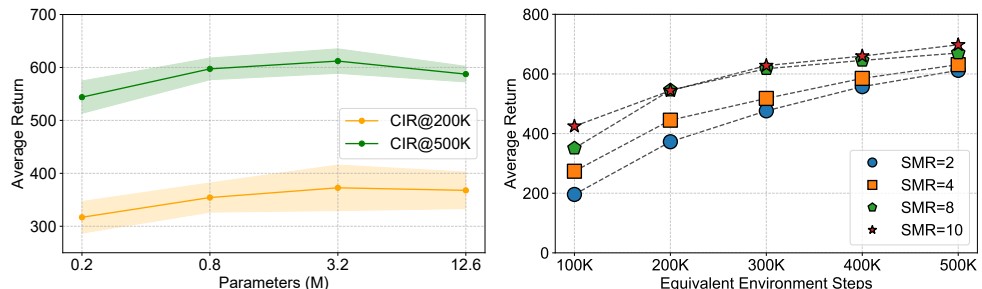

Figure 6: **Scaling results of CIR.** We report the scaling results on network capacity (**left**) and the SMR ratio (**right**). We use the aggregated average return for performance comparison.

that the representations in downstream layers are stabilized. Layer normalization also serves as a regularization term (as required in Theorem 4.5). Both of them are vital for CIR. Also, either removing the skip connection module or replacing the `Tanh` function with sigmoid, softmax, or layer normalization can incur performance drop. These verify our design choice on network components.

**On algorithmic components.** The results show that adopting CDQ for value estimate causes significant performance decrease. This validates our claim in Section 5.3 and stresses the necessity of convex Q-learning. We observe that **UTD** achieves similar performance as CIR on medium tasks but its performance on hard tasks is worse than CIR, demonstrating the superiority of SMR over UTD.

## 6.4 SCALING RESULTS

We now investigate the scaling capability of CIR, and adopt 6 DMC medium tasks, 3 DMC hard tasks for experiments. Since medium tasks are run for 500K steps while hard tasks are run for 1M steps, we define *equivalent environment step $k$* that aligns the actual environment step of medium and hard tasks. Reporting average returns under equivalent environment step $k$ means that we aggregate returns of medium tasks at environment step $k$ and hard tasks at environment step $2k$. We denote CIR@$k$ as the (aggregated) return of CIR at $k$-th step in medium tasks and $2k$-th step in hard tasks. For example, CIR@200K means the aggregated return of DMC medium tasks at 200K steps and hard tasks at 400K steps. The empirical results are outlined in Figure 6.

**Scaling the network size.** We scale the width of the CIR critic networks and the actor network by varying the size of the hidden dimension in $\{128, 256, 512, 1024\}$. The results in Figure 6 (left) illustrate that CIR generally benefits from scaling the network capacity across different equivalent environment steps. The performance of CIR grows slowly upon reaching a certain amount of parameters, which matches the reported scaling results in (Nauman et al., 2024; Lee et al., 2025a).

**Scaling the SMR ratio.** We then investigate whether CIR can benefit from scaling computation time by measuring the performance of CIR across different SMR ratios (from 2 to 10). Figure 6 (right) demonstrates that CIR consistently enjoys performance improvement by increasing the SMR ratio under various equivalent environment steps, especially at the initial training stage.

## 7 CONCLUSION AND FUTURE OPPORTUNITIES

Our key contribution in this work is the CIR framework, which aims at enhancing the sample efficiency from the perspective of initial representation constraint. CIR introduces three novel components, (i) restricting initial representations by leveraging the `Tanh` function, layer normalization, and AvgRNorm; (ii) the skip connection technique that offers a linear pathway from the low-level features to deep abstract representation; (iii) the convex Q-learning approach that returns optimistic and flexible value estimate. We provide theoretical guarantees for our method under linear function approximation. Moreover, empirical evaluations on the DMC suite, HumanoidBench and ODRL tasks show that CIR exhibits competitive or even better performance against recent strong baselines.

**Opportunities.** We believe that this work offers a valuable attempt at improving sample efficiency by constraining initial representations, offering opportunities for proposing better representation stabilization methods (e.g., combining various normalization tricks) or designing better architectures.

ETHICS STATEMENT

This research does not involve human subjects, animal experiments, or sensitive datasets, nor does it include applications that may pose ethical risks. We believe that this study does not raise any ethical concerns.

REPRODUCIBILITY STATEMENT

We have made efforts to ensure that our work is reproducible. We provide the source code of our algorithm in the supplementary materials. Additionally, we provide the detailed hyperparameter setup for CIR in Appendix D, and the details on benchmark environments in Appendix C.

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

# A  MISSING PROOFS

In this section, we provide missing proofs for theorems in the main text. For better readability, we restate all theorems.

## A.1  PROOF OF THEOREM 4.2

**Theorem A.1** (Linear Independence). *Suppose that $W = c \cdot I$, where $c \in \mathbb{R}$ is a sufficient small positive number, $I$ is the identity matrix, then if basis functions $\{\phi_1, \ldots, \phi_S\}$ are linearly independent, then the transformed basis functions $\{\tanh(W\phi_1), \ldots, \tanh(W\phi_S)\}$ are also linearly independent.*

*Proof.* Since we set $W = c \cdot I$, we have $\tanh(W\phi_j) = \tanh(c \cdot \phi_j), \forall j \in \{1, 2, \ldots, S\}$. Based on the assumption that the basis functions $\{\phi_1, \ldots, \phi_S\}$ are linearly independent, the only solution to the equation

$$\sum_{i=1}^{S} a_i \phi_i = 0 \tag{11}$$

is $a_1 = a_2 = \cdots = a_S = 0$. When $c$ is sufficiently small, $c \cdot \phi_i$ is also sufficiently small, $\forall i \in \{1, 2, \ldots, S\}$. By using Taylor expansion of $\tanh(x)$ around $x = 0$, we have

$$\tanh(x) = x - \frac{x^3}{3} + \frac{2x^5}{15} - \cdots = x + \mathcal{O}(x^3). \tag{12}$$

Thus, for $c \cdot \phi_i \approx \mathbf{0}$ where $\mathbf{0}$ is $d$-dimensional zero matrix, we have

$$\tanh(c \cdot \phi_i) = c \cdot \phi_i + \mathcal{O}(c^3 \|\phi_i\|^3). \tag{13}$$

Suppose there exists a non-zero coefficient vector $\mathbf{a} = (a_1, a_2, \ldots, a_S)$ such that

$$\sum_{i=1}^{S} a_i \tanh(c \cdot \phi_i) = 0. \tag{14}$$

Substituting the Taylor expansion, we get

$$\sum_{i=1}^{S} a_i \left( c \cdot \phi_i + \mathcal{O}(c^3 \|\phi_i\|^3) \right) = 0, \tag{15}$$

which simplifies to

$$\sum_{i=1}^{S} a_i \phi_i + \mathcal{O}(c^2) = 0. \tag{16}$$

As $c \to 0$, the remainder term $\mathcal{O}(c^2) \to 0$. By the continuity of linear combinations, we obtain

$$\sum_{i=1}^{S} a_i \phi_i = 0, \tag{17}$$

which contradicts the linear independence of $\{\phi_i\}$. Thus, no such non-zero vector $\mathbf{a}$ can exist for sufficiently small $c$. This indicates that the transformed basis functions $\{\tanh(W\phi_1), \ldots, \tanh(W\phi_S)\}$ are also linearly independent. $\square$

## A.2  PROOF OF THEOREM 4.3

**Theorem A.2** (Variance Reduction). *Given any state $s$, we assume the feature $\phi(s)$ follows a distribution with $\mathbb{E}[\phi(s)] = 0$. Consider per-transition least square TD (LSTD) objective $\mathcal{L} = \frac{1}{2}[(r + \gamma V(s'; \theta) - V(s; \theta))^2]$, then if $V(s; \theta) = \phi(s)^\mathsf{T}\theta$, the variance of the semi-gradient term satisfies:*

$$\|\mathrm{Var}(\nabla_\theta \mathcal{L})\| \le (r_{\max} + (1 + \gamma)\|\theta\|\Lambda)^2 \cdot \|\phi(s)\|^2, \tag{18}$$

*where $\Lambda = \max\{\|\phi(s)\|, \|\phi(s')\|\}$. While if $V(s; \theta) = \tanh(W\phi(s))^\mathsf{T}\theta$, we have*

$$\|\mathrm{Var}(\nabla_\theta \mathcal{L})\| \le (r_{\max} + (1 + \gamma)\|\theta\|\lambda_W \Lambda)^2 \cdot \|\tanh(W\phi(s))\|^2, \tag{19}$$

*where $\lambda_W$ is the maximum eigenvalue of $W$.*

*Proof.* Given the objective function

$$\mathcal{L} = \frac{1}{2}[(r + \gamma V(s'; \theta) - V(s; \theta))^2], \tag{20}$$

it is easy to find its semi-gradient term,

$$\nabla \mathcal{L} = (r + \gamma V(s'; \theta) - V(s; \theta))\nabla V(s; \theta). \tag{21}$$

If $V(s; \theta) = \phi(s)^\mathsf{T}\theta$, we have

$$\nabla_\theta \mathcal{L} = (r + \gamma \phi(s')^\mathsf{T}\theta - \phi(s)^\mathsf{T}\theta)\phi(s). \tag{22}$$

Recall that $\mathrm{Var}(X) = \mathbb{E}[X^2] - (\mathbb{E}X)^2$, we notice that $\phi(s)^\mathsf{T}\theta, \phi(s')^\mathsf{T}\theta$ are scalar values, by using the assumption, it is easy to find that

$$\mathbb{E}[\nabla_\theta \mathcal{L}] = \mathbb{E}[(r + \gamma \phi(s')^\mathsf{T}\theta - \phi(s)^\mathsf{T}\theta)\phi(s)] = (r + \gamma \phi(s')^\mathsf{T}\theta - \phi(s)^\mathsf{T}\theta)\mathbb{E}[\phi(s)] = 0. \tag{23}$$

Therefore, we have

$$\mathrm{Var}(\nabla_\theta \mathcal{L}) = \mathbb{E}[\nabla_\theta \mathcal{L}]^2 = [r + \gamma \phi(s')^\mathsf{T}\theta - \phi(s)^\mathsf{T}\theta]^2 \mathbb{E}[\phi(s)\phi(s)^\mathsf{T}]. \tag{24}$$

Then,

$$\|\mathrm{Var}(\nabla_\theta \mathcal{L})\| = \|[r + \gamma \phi(s')^\mathsf{T}\theta - \phi(s)^\mathsf{T}\theta]^2 \mathbb{E}[\phi(s)\phi(s)^\mathsf{T}]\| \tag{25}$$

$$\leq (r_{\max} + \gamma\|\phi(s')\|\|\theta\| + \|\phi(s)\|\|\theta\|)^2\|\phi(s)\|^2 \tag{26}$$

$$\leq (r_{\max} + (1 + \gamma)\|\theta\|\Lambda)^2\|\phi(s)\|^2, \tag{27}$$

where $\Lambda = \max\{\|\phi(s)\|, \|\phi(s')\|\}$. While if $V(s; \theta) = \tanh(W\phi(s))^\mathsf{T}\theta$, where $W$ is a positive definite diagonal matrix, the semi-gradient gives

$$\nabla_\theta \mathcal{L} = \underbrace{(r + \gamma\tanh(W\phi(s'))^\mathsf{T}\theta - \tanh(W\phi(s))^\mathsf{T}\theta)}_{:=\delta}\tanh(W\phi(s)). \tag{28}$$

We first bound $\mathbb{E}[\nabla_\theta \mathcal{L}]^2$ term,

$$\mathbb{E}[\nabla_\theta \mathcal{L}]^2 = \|[r + \gamma\tanh(W\phi(s'))^\mathsf{T}\theta - \tanh(W\phi(s))^\mathsf{T}\theta]^2 \mathbb{E}[\tanh(W\phi(s))\tanh(W\phi(s))^\mathsf{T}]\| \tag{29}$$

$$\leq (r_{\max} + \gamma\|\tanh(W\phi(s'))\|\|\theta\| + \|\tanh(W\phi(s))\|\|\theta\|)^2 \cdot \|\tanh(W\phi(s))\|^2 \tag{30}$$

$$= (r_{\max} + (1 + \gamma)\|\theta\|\lambda_W\Lambda)^2 \cdot \|\tanh(W\phi(s))\|^2, \tag{31}$$

where the last inequality holds due to $\|\tanh(W\phi(s))\| \leq \|W\phi(s)\| \leq \lambda_W\|\phi(s)\| \leq \lambda_W\Lambda$.

Unfortunately, $\mathbb{E}[\phi(s)] = 0$ does not necessarily guarantee that $\mathbb{E}[\tanh(W\phi(s))] = 0$. Nevertheless, we have $\|\tanh(W\phi(s))\| \leq \sqrt{d}$ since $|\tanh(x)| \leq 1$. We then have the variance term,

$$\|\mathrm{Var}(\nabla_\theta \mathcal{L})\| = \mathbb{E}[\nabla_\theta \mathcal{L}]^2 - (\mathbb{E}[\nabla_\theta \mathcal{L}])^2 \tag{32}$$

$$= \mathbb{E}[\nabla_\theta \mathcal{L}]^2 - \delta^2 \cdot (\mathbb{E}[\tanh(W\phi(s))])^2 \tag{33}$$

$$\leq (r_{\max} + (1 + \gamma)\|\theta\|\lambda_W\Lambda)^2 \cdot \|\tanh(W\phi(s))\|^2 - \mathcal{O}(d) \tag{34}$$

$$\leq (r_{\max} + (1 + \gamma)\|\theta\|\lambda_W\Lambda)^2 \cdot \|\tanh(W\phi(s))\|^2. \tag{35}$$

These conclude the proof. □

### A.3 PROOF OF THEOREM 4.4

**Theorem A.3** (Convergence under TD(0))**.** *Consider the TD(0) update rule for the value function* $V(s; \theta) = \tanh(W\phi(s))^\mathsf{T}\theta$. *Suppose we project the parameters* $\theta$ *onto bounded sets* $\Theta = \{\theta \mid \|\theta\|_2 \leq C_\theta\}$ *after each update. If the step sizes satisfy* $\sum \alpha_t = \infty$, $\sum \alpha_t^2 < \infty$, *then the projected TD(0) algorithm converges almost surely to a fixed point* $\theta^*$.

*Proof.* Denote $\tilde{\phi}(s) = \tanh(W\phi(s))$, then the value function becomes $V(s; \theta) = \tilde{\phi}^\mathsf{T}\theta$, and the new features $\tilde{\phi}(s)$ satisfy $\|\tilde{\phi}(s)\| = \|\tanh(W\phi(s))\| \leq \sqrt{d} < \infty$. Therefore, the problem degenerates into the convergence of TD(0) under the formulation $V(s; \theta) = \tilde{\phi}^\mathsf{T}\theta$, which is a well-known result based on Theorem 1 in (Tsitsiklis & Van Roy, 1996). The resulting fixed point $\theta^*$ is the fixed point of the projected Bellman equation,

$$\tanh(W\phi(s))^\mathsf{T}\theta = \Pi_\nu \mathcal{T}(\tanh(W\phi(s))^\mathsf{T}\theta), \tag{36}$$

where $\Pi_\nu(x) = \arg\min_{z \in \tanh(W\phi(s))^\mathsf{T}\theta} \|z - x\|_\nu^2$, $\mathcal{T}$ is the Bellman operator, $\nu$ is the stationary distribution of the Markov chain. □

## A.4    PROOF OF THEOREM 4.5

**Theorem A.4** (Global Convergence under Regularization). *Given the regularized loss $\mathcal{L}(\theta) = \mathbb{E}\left[\left(r + \gamma \tanh(W\phi(s'))^\mathsf{T}\theta - \tanh(W\phi(s))^\mathsf{T}\theta\right)^2\right] + \lambda_\theta\|\theta\|_2^2$, assume that (a) the step size $\alpha$ is a constant; (b) the gradient of the loss $\mathcal{L}(\theta)$ is $\beta$-smooth, i.e., $\|\nabla\mathcal{L}(\theta_1) - \nabla\mathcal{L}(\theta_2)\| \leq \beta\|\theta_1 - \theta_2\|$; (c) the regularization parameter satisfies $0 < \lambda_\theta < \frac{\beta}{2}$. Then, gradient descent with step size $\alpha \leq 1/\beta$ converges linearly to a unique global minimum $\theta^*$.*

*Proof.* The loss function combines the Bellman error and regularization

$$\mathcal{L}(\theta) = \underbrace{\mathbb{E}\left[\left(r + \gamma\tanh(W\phi(s'))^\mathsf{T}\theta - \tanh(W\phi(s))^\mathsf{T}\theta\right)^2\right]}_{\text{Quadratic in }\theta} + \lambda_\theta\|\theta\|^2.$$

Note that though $\tanh(W\phi(s))$ is nonlinear in $\phi(s)$, $V(s;\theta)$ is linear in $\theta$ and hence the Bellman error term is quadratic in $\theta$. We then expand the Bellman error term:

$$\mathbb{E}\left[\left(r + (\gamma\tanh(W\phi(s'))^\mathsf{T}\theta - \tanh(W\phi(s))^\mathsf{T}\theta\right)^2\right] = \theta^\mathsf{T}A\theta - 2\theta^\mathsf{T}b + c, \tag{37}$$

where

$$A = \mathbb{E}\left[(\gamma\tanh(W\phi(s')) - \tanh(W\phi(s)))(\gamma\tanh(W\phi(s')) - \tanh(W\phi(s)))^\mathsf{T}\right], \tag{38}$$

$$b = \mathbb{E}\left[r(\gamma\tanh(W\phi(s')) - \tanh(W\phi(s)))\right] \tag{39}$$

$$c = \mathbb{E}[r^2]. \tag{40}$$

Adding the L2-regularization term, the total loss becomes:

$$\mathcal{L}(\theta) = \theta^\mathsf{T}(A + \lambda_\theta I)\theta - 2\theta^\mathsf{T}b + c. \tag{41}$$

Note that the Hessian matrix $H = 2(A + \lambda_\theta I) \succeq 2\lambda_\theta I$. Since $\lambda_\theta > 0$, the matrix $A + \lambda_\theta I$ is positive definite, ensuring that $\mathcal{L}(\theta)$ is $2\lambda_\theta$-*strongly convex*. It is well-known that a strongly convex function has exactly one global minimum. The optimal $\theta^*$ satisfies:

$$\nabla_\theta\mathcal{L}(\theta^*) = 0 \implies (A + \lambda_\theta I)\theta^* = b.$$

Solving this linear system yields $\theta^* = (A + \lambda_\theta I)^{-1}b$. Then, we show that using gradient descent ensures linear convergence to the global minimum $\theta^*$.

We expand $\mathcal{L}(\theta_2)$ around $\theta_1$ using Taylor's theorem:

$$\mathcal{L}(\theta_2) = \mathcal{L}(\theta_1) + \nabla f(\theta_1)^\mathsf{T}(\theta_2 - \theta_1) + \frac{1}{2}(\theta_2 - \theta_1)^\mathsf{T}\nabla^2 f(\xi)(\theta_2 - \theta_1), \tag{42}$$

where $\xi$ lies on the line segment between $\theta_1$ and $\theta_2$. Recall that the gradient of the loss function is $\beta$-smooth, i.e.,

$$\|\nabla\mathcal{L}(\theta_2) - \nabla\mathcal{L}(\theta_1)\| \leq \beta\|\theta_2 - \theta_1\|. \tag{43}$$

It indicates that the largest eigenvalue of $\nabla^2\mathcal{L}(\xi)$ is bounded by $\beta$. Thus:

$$\frac{1}{2}(\theta_2 - \theta_1)^\mathsf{T}\nabla^2\mathcal{L}(\xi)(\theta_2 - \theta_1) \leq \frac{\beta}{2}\|\theta_2 - \theta_1\|^2. \tag{44}$$

Substituting back induces:

$$\mathcal{L}(\theta_2) \leq \mathcal{L}(\theta_1) + \nabla\mathcal{L}(\theta_1)^\mathsf{T}(\theta_2 - \theta_1) + \frac{\beta}{2}\|\theta_2 - \theta_1\|^2. \tag{45}$$

Recall that $\mathcal{L}$ is $2\lambda_\theta$-strongly convex, based on its definition, we have for all $\theta_1, \theta_2$:

$$\mathcal{L}(\theta_2) \geq \mathcal{L}(\theta_1) + \nabla\mathcal{L}(\theta_1)^\mathsf{T}(\theta_2 - \theta_1) + \lambda_\theta\|\theta_2 - \theta_1\|^2. \tag{46}$$

Apply strong convexity at $\theta_1 = \theta^*$, we have

$$\mathcal{L}(\theta) \geq \mathcal{L}(\theta^*) + \underbrace{\nabla\mathcal{L}(\theta^*)^\mathsf{T}(\theta - \theta^*)}_{=0} + \lambda_\theta\|\theta - \theta^*\|^2. \tag{47}$$

Denote $\bar{\lambda} := 2\lambda_\theta$, and we have

$$\mathcal{L}(\theta) - \mathcal{L}(\theta^*) \geq \frac{\bar{\lambda}}{2} \|\theta - \theta^*\|^2. \tag{48}$$

Next, we use the Polyak-Łojasiewicz (PL) inequality (Nesterov, 2013; Karimi et al., 2016), which relates the gradient norm to the function suboptimality,

$$\|\nabla\mathcal{L}(\theta)\|^2 \geq 2\bar{\lambda}\left(\mathcal{L}(\theta) - \mathcal{L}(\theta^*)\right). \tag{49}$$

Now we consider the gradient descent with the following form:

$$\theta_{t+1} = \theta_t - \alpha\nabla\mathcal{L}(\theta_t), \tag{50}$$

where $\alpha$ is the step size. Setting $\theta_2 = \theta_{t+1}$ and $\theta_1 = \theta_t$ in Equation 45 and we have

$$\mathcal{L}(\theta_{t+1}) \leq \mathcal{L}(\theta_t) - \alpha\|\nabla\mathcal{L}(\theta_t)\|^2 + \frac{\beta\alpha^2}{2}\|\nabla\mathcal{L}(\theta_t)\|^2 \tag{51}$$

$$= \mathcal{L}(\theta_t) - \alpha\left(1 - \frac{\beta\alpha}{2}\right)\|\nabla\mathcal{L}(\theta_t)\|^2. \tag{52}$$

For $\alpha \leq \frac{1}{\beta}$, the term $\left(1 - \frac{\beta\alpha}{2}\right) \geq \frac{1}{2}$, hence we have

$$\mathcal{L}(\theta_{t+1}) \leq \mathcal{L}(\theta_t) - \frac{\alpha}{2}\|\nabla\mathcal{L}(\theta_t)\|^2. \tag{53}$$

By using Equation 49, we have that $\|\nabla\mathcal{L}(\theta_t)\|^2 \geq \frac{\bar{\lambda}}{2}\left(\mathcal{L}(\theta_t) - \mathcal{L}(\theta^*)\right)$ and combining it with Equation 53, we have

$$\mathcal{L}(\theta_{t+1}) \leq \mathcal{L}(\theta_t) - \frac{\alpha}{2} \cdot 2\bar{\lambda}\left(\mathcal{L}(\theta_t) - \mathcal{L}(\theta^*)\right). \tag{54}$$

This implies that

$$\mathcal{L}(\theta_{t+1}) - \mathcal{L}(\theta^*) \leq \left(1 - \alpha\bar{\lambda}\right)\left(\mathcal{L}(\theta_t) - \mathcal{L}(\theta^*)\right). \tag{55}$$

This indicates a linear convergence rate with contraction factor $1 - \alpha\bar{\lambda} = 1 - 2\alpha\lambda_\theta$. For $\alpha = \frac{1}{\beta}$:

$$\mathcal{L}(\theta_t) - \mathcal{L}(\theta^*) \leq \left(1 - \frac{2\lambda_\theta}{\beta}\right)^t \left(\mathcal{L}(\theta_0) - \mathcal{L}(\theta^*)\right).$$

Since $0 < \lambda_\theta < \frac{\beta}{2}$, the term $\left(1 - \frac{2\lambda_\theta}{\beta}\right) < 1$, ensuring exponential decay. Hence, we conclude that the weight $\theta$ converges to a global minimum $\theta^*$ linearly. $\square$

## A.5   PROOF OF THEOREM 5.1

The proof of Theorem 5.1 quite resembles the proof in TD3 (Fujimoto et al., 2018). We show it in the finite MDP setting, where we maintain two tabular value estimates $Q^A, Q^B$. At each timestep $t$, we select action $a^*$ via $a^* = \arg\max_a Q_t^A(s, a)$ and calculate the target value of the convex Q-learning,

$$y = r + \gamma\left(\lambda \times \min(Q^A(s', a^*), Q^B(s', a^*)) + (1 - \lambda) \times \max(Q^A(s', a^*), Q^B(s', a^*))\right). \tag{56}$$

Then, we update the value estimates by using the vanilla Q-learning update formula:

$$\begin{aligned} Q_{t+1}^A(s, a) &\leftarrow Q_t^A(s, a) + \alpha_t(y - Q_t^A(s, a)), \\ Q_{t+1}^B(s, a) &\leftarrow Q_t^B(s, a) + \alpha_t(y - Q_t^B(s, a)), \end{aligned} \tag{57}$$

where $\alpha_t$ is the learning rate. We also need to use a well-known lemma that is adopted in (Fujimoto et al., 2018) [2].

---

[2]The proof can be found in https://link.springer.com/content/pdf/10.1023/a:1007678930559.pdf

**Lemma A.5.** *Consider a stochastic process $(\zeta_t, \Delta_t, F_t), t \geq 0$ where $\zeta_t, \Delta_t, F_t : X \mapsto \mathbb{R}$ satisfy the equation:*

$$\Delta_{t+1}(x_t) = (1 - \zeta_t(x_t))\Delta_t(x_t) + \zeta_t(x_t)F_t(x_t), \tag{58}$$

*where $x_t \in X$ and $t = 0, 1, 2, \ldots$. Let $P_t$ be a sequence of increasing $\sigma$-fields such that $\zeta_0$ and $\Delta_0$ are $P_0$-measurable and $\zeta_t, \Delta_t$ and $F_{t-1}$ are $P_t$-measurable, $t = 1, 2, \ldots$. Assume the following conditions hold: (1) The set $X$ is finite; (2) $\zeta_t(x_t) \in [0, 1]$, $\sum_t \zeta_t(x_t) = \infty$, $\sum_t (\zeta_t(x_t))^2 < \infty$ with probability 1 and $\forall x \neq x_t : \zeta_t(x_t) = 0$; (3) $\|\mathbb{E}[F_t|P_t]\| \leq \kappa \|\Delta_t\| + c_t$, where $\|\cdot\|$ denotes maximum norm, $\kappa \in [0, 1)$ and $c_t$ converges to 0 with probability 1; (4) $Var[F_t(x_t)|P_t] \leq C(1 + \|\Delta_t\|)^2$, where $C$ is some constant. Then $\Delta_t$ converges to 0 with probability 1.*

Then we are ready to show the following theorem.

**Theorem A.6** (Theorem 5.1 restated). *Given the following conditions: (1) each state-action pair is sampled an infinite number of times; (2) the MDP is finite; (3) $\gamma \in [0, 1)$; (4) Q values are stored in a look-up table; (5) the learning rates satisfy $\alpha_t(s, a) \in [0, 1]$, $\sum_t \alpha_t(s, a) = \infty$, $\sum_t (\alpha_t(s, a))^2 < \infty$ with probability 1 and $\alpha_t(s, a) = 0, \forall(s, a) \neq (s_t, a_t)$; (6) $Var[r(s, a)] < \infty, \forall s, a$; (7) the Q-values receive an infinite number of updates, then for any $\lambda \in (0, 1]$, the Convex Q-learning converges to the optimal value function $Q^*$ as defined by the Bellman optimality equation with probability 1.*

*Proof.* To apply Lemma A.5, we set $P_t = \{Q_0^A, Q_0^B, s_0, a_0, \ldots, s_t, a_t\}$, $X = \mathcal{S} \times \mathcal{A}$, $\zeta_t = \alpha_t$, and $\Delta_t = Q_t^A - Q^*$. Notice that the conditions (1) and (2) of Lemma A.5 hold due to the assumed conditions (2) and (5) in our theorem, respectively. The condition (4) of the lemma holds by the condition (6) in our theorem. Denote $a^* = \arg\max_a Q^A(s_{t+1}, a)$, we have

$$
\begin{aligned}
&\Delta_{t+1}(s_t, a_t)\\
&= Q_{t+1}^A(s_t, a_t) - Q^*(s_t, a_t)\\
&= (1 - \alpha_t)\underbrace{(Q_t^A(s_t, a_t) - Q^*(s_t, a_t))}_{=\Delta_t} + \alpha_t(y - Q^*(s_t, a_t))\\
&= (1 - \alpha_t)\Delta_t + \\
&\alpha_t \underbrace{(r_t + \gamma[\lambda \min(Q_t^A(s_{t+1}, a^*), Q_t^B(s_{t+1}, a^*)) + (1 - \lambda)\max(Q_t^A(s_{t+1}, a^*), Q_t^B(s_{t+1}, a^*))] - Q^*(s_t, a_t))}_{:=F_t(s_t, a_t)}\\
&= (1 - \alpha_t)\Delta_t + \alpha_t F_t(s_t, a_t).
\end{aligned}
\tag{59}
$$

For the defined $F_t(s_t, a_t)$, we have

$$
\begin{aligned}
&F_t(s_t, a_t)\\
&= r_t + \gamma[\lambda \min(Q_t^A(s_{t+1}, a^*), Q_t^B(s_{t+1}, a^*))\\
&\qquad + (1 - \lambda)\max(Q_t^A(s_{t+1}, a^*), Q_t^B(s_{t+1}, a^*))] - Q^*(s_t, a_t)\\
&= r_t + \gamma[\lambda \min(Q_t^A(s_{t+1}, a^*), Q_t^B(s_{t+1}, a^*))\\
&\qquad + (1 - \lambda)\max(Q_t^A(s_{t+1}, a^*), Q_t^B(s_{t+1}, a^*))] - Q^*(s_t, a_t) + \gamma Q_t^A(s_{t+1}, a^*) - \gamma Q_t^A(s_{t+1}, a^*)\\
&= F_t^Q(s_t, a_t) + c_t,
\end{aligned}
\tag{60}
$$

where we define $F_t^Q(s_t, a_t)$ and $c_t$ below,

$$
\begin{aligned}
F_t^Q(s_t, a_t) &= r_t + \gamma Q_t^A(s_{t+1}, a^*) - Q^*(s_t, a_t),\\
c_t &= \gamma[\lambda \min(Q_t^A(s_{t+1}, a^*), Q_t^B(s_{t+1}, a^*)) + (1 - \lambda)\max(Q_t^A(s_{t+1}, a^*), Q_t^B(s_{t+1}, a^*))]\\
&\qquad - \gamma Q_t^A(s_{t+1}, a^*).
\end{aligned}
\tag{61}
$$

Note that $\mathbb{E}[F_t^Q|P_t] \leq \gamma \|\Delta_t\|$ is a well-known result. It remains to show that $c_t$ converges to 0 with probability 1. Denote $\Delta_t^{BA}(s_t, a_t) = Q_t^B(s_t, a_t) - Q_t^A(s_t, a_t)$, then $c_t$ converges to 0 if $\Delta_t^{BA}(s_t, a_t)$ converges to 0, which holds for any $\lambda \in (0, 1]$. We then aim at showing that $\Delta_t^{BA}$ converges to 0

with probability 1. Based on the definition and the update rule of Q-learning, we have

$$
\begin{aligned}
\Delta_{t+1}^{BA}(s_t, a_t) &= Q_{t+1}^{B}(s_t, a_t) - Q_{t+1}^{A}(s_t, a_t) \\
&= Q_t^{B}(s_t, a_t) + \alpha_t(y - Q_t^{B}(s_t, a_t)) - Q_t^{A}(s_t, a_t) - \alpha_t(y - Q_t^{A}(s_t, a_t)) \\
&= (1 - \alpha_t)(Q_t^{B}(s_t, a_t) - Q_t^{A}(s_t, a_t)) \\
&= (1 - \alpha_t)\Delta_t^{BA}(s_t, a_t).
\end{aligned}
\tag{62}
$$

We then conclude that $\Delta_t^{BA}$ converges to 0, indicating that the condition (3) of Lemma A.5 is satisfied. Then, by using Lemma A.5, we have $Q^A$ converges to $Q^*$ with probability 1. Similarly, we get the convergence of $Q^B$ by defining $\Delta_t = Q_t^B - Q^*$ and following the same procedure above. Combining these results, the proof is completed. □

## B  PSEUDO-CODES FOR CIR

We present the detailed pseudo-code for CIR in Algorithm 1.

---

**Algorithm 1** Constrained Initial Representations (CIR)

---

1: Initialize critic networks $Q_{\theta_1}, Q_{\theta_2}$ and actor network $\pi_\phi$ with random parameters
2: Initialize target networks $\theta_1' \leftarrow \theta_1, \theta_2' \leftarrow \theta_2$ and replay buffer $\mathcal{D} = \{\}$
3: Initialize temperature coefficient $\alpha$, target update rate $\tau$, number of interaction steps $T$
4: Set value coefficient $\lambda$, normalization parameter $c$, SMR ratio $M$
5: **for** $t = 1$ to $T$ **do**
6:   Take one action $a_t \sim \pi_\phi(\cdot|s_t)$ and observe reward $r_t$, new state $s_{t+1}'$
7:   Store transitions in the replay buffer, i.e., $\mathcal{D} \leftarrow \mathcal{D} \bigcup \{(s_t, a_t, r_t, s_{t+1}')\}$
8:   Sample a mini-batch $B = \{(s, a, r, s')\} \sim \mathcal{D}$
9:   **for** $m = 1$ to $M$ **do**
10:    Compute the $Q$ target using the convex Q-learning approach:

$$y = r + \gamma \left( \lambda \times \min_{i \in \{1,2\}} Q_{\theta_i'}(s', a') + (1 - \lambda) \times \max_{i \in \{1,2\}} Q_{\theta_i'}(s', a') - \alpha \log \pi_\phi(a'|s') \right),$$

    where $a' \sim \pi_\phi(\cdot|s')$
11:    Update $\theta_i$ with gradient descent using $\nabla_{\theta_i} \frac{1}{|B|} \sum_{(s,a,r,s') \sim B} (Q_{\theta_i}(s,a) - y)^2$
12:    Update target networks: $\theta_i' \leftarrow \tau\theta_i + (1 - \tau)\theta_i'$
13:    Update actor $\phi$ with $\nabla_\phi \frac{1}{|B|} \sum_{s \in B} \left( \frac{1}{2} \sum_{j=1}^{2} Q_{\theta_j}(s, \tilde{a}) - \alpha \log \pi_\phi(\tilde{a}|s) \right), \tilde{a} \sim \pi_\phi(\cdot|s)$
14:   **end for**
15: **end for**

---

We further present below the detailed comparison of CIR against prior methods. Different from prior methods, we explore a novel way of constraining initial representations to improve sample efficiency, where CIR involves different architecture components and algorithm components.

Table 2: **Architecture and algorithmic comparison between CIR and prior methods.**

| Algorithm | Architecture Components | Algorithm Components |
|---|---|---|
| BRO | LayerNorm, residual connection, weight decay, parameter reset | Distributional Q-learning, Huber loss, dual policy exploration, UTD |
| SimBa | RSNorm, pre-layer normalization, post-layer normalization, residual connection, weight decay | single/double Q-learning, UTD |
| **CIR (ours)** | Tanh, AvgRNorm, LayerNorm, skip connection | SMR, Convex Q-learning |

## C  Environment Details

In this part, we give detailed environment descriptions for selected benchmarks. We mainly consider DMC suite (Tassa et al., 2018), HumanoidBench (Sferrazza et al., 2024), and ODRL (Lyu et al., 2024c). The visualization results of selected tasks from these benchmarks can be found in Figure 3.

**DeepMind Control suite.** DeepMind Control suite (Tassa et al., 2018) (DMC suite) is a popular continuous control benchmark in RL, which utilizes proprioceptive states as the observation space. It contains a wide range of continuous control tasks with varying complexities, involving both low-dimensional tasks (the state dimension can give only 3, the action dimension can give only 1) and high-dimensional tasks (the state dimension can give 223, the action dimension can give 38). The total reward of each episode in DMC suite is limited to 1000, making it comparatively easy to aggregate results. We choose 27 DMC suite tasks for evaluation, including 20 DMC suite easy & medium tasks, and 7 hard tasks (`humanoid` and `dog` tasks). We show in Table 3 the detailed information (state space dimension and action space dimension) of the adopted DMC easy & medium tasks. We also present the detailed task name, state space information, and action space information of DMC hard tasks in Table 4. For all DMC suite tasks, we use the average episode return as the performance metric. We run DMC easy & medium tasks for 500K environment steps and hard tasks for 1M environment steps. The action repeat for all DMC tasks is set to be 2.

Table 3: **Detailed state and action space dimensions for DMC easy and medium tasks.**

| Task | Observation dimension | Action dimension |
|---|---|---|
| Acrobot Swingup | 6 | 1 |
| Cartpole Balance | 5 | 1 |
| Cartpole Balance Sparse | 5 | 1 |
| Cartpole Swingup | 5 | 1 |
| Cartpole Swingup Sparse | 5 | 1 |
| Cheetah Run | 17 | 6 |
| Finger Spin | 9 | 2 |
| Finger Turn Easy | 12 | 2 |
| Finger Turn Hard | 12 | 2 |
| Fish Swim | 24 | 5 |
| Hopper Hop | 15 | 4 |
| Hopper Stand | 15 | 4 |
| Pendulum Swingup | 3 | 1 |
| Quadruped Run | 78 | 12 |
| Quadruped Walk | 78 | 12 |
| Reacher Easy | 6 | 2 |
| Reacher Hard | 6 | 2 |
| Walker Run | 24 | 6 |
| Walker Stand | 24 | 6 |
| Walker Walk | 24 | 6 |

Table 4: **Detailed state and action space dimensions for DMC hard tasks.**

| Task | Observation dimension | Action dimension |
|---|---|---|
| Dog Run | 223 | 38 |
| Dog Trot | 223 | 38 |
| Dog Stand | 223 | 38 |
| Dog Walk | 223 | 38 |
| Humanoid Run | 67 | 24 |
| Humanoid Stand | 67 | 24 |
| Humanoid Walk | 67 | 24 |

**HumanoidBench.** HumanoidBench (Sferrazza et al., 2024) is a high-dimensional simulated robot learning benchmark built upon the Unitree H1 humanoid robot. HumanoidBench provides an acces-

sible, fast, safe, and inexpensive testbed for the field of robot learning research. It demonstrates a variety of challenges in learning for autonomous humanoid robots, including the intricate control of robots with complex dynamics, sophisticated coordination among various body parts, and handling long-horizon complex tasks. It features a variety of challenging whole-body manipulation tasks with dexterous hands and locomotion tasks. HumanoidBench offers a promising platform for facilitating prompt verification of algorithms and ideas in humanoid robots. Following SimBa (Lee et al., 2025a), we consider 14 locomotion tasks without dexterous hands. The observation dimension of these tasks ranges from 51 to 77, and the action space dimension of these tasks gives 19. We run HumanoidBench tasks for 2M environment steps with action repeat 2. Note that HumanoidBench tasks have quite distinct average episode returns for different tasks. To better aggregate results, we use *normalized return* as the metric, which normalizes the performance score of the agent by their corresponding task success score. Denote the task success score of task $i$ from HumanoidBench as $S_i$, the average episode return of the agent in the task $i$ as $R_i$, then the normalized return $NS_i$ for task $i$ is computed by:

$$NS_i = \frac{R_i}{S_i} \times 1000. \tag{63}$$

We summarize the information on the state dimensions, action dimensions, and the corresponding task success scores of our selected tasks from HumanoidBench in Table 5.

Table 5: **Detailed state, action space dimensions and task success scores for HumanoidBench tasks.**

| Task | Observation dimension | Action dimension | Task success score |
|------|----------------------|------------------|--------------------|
| Balance Hard | 77 | 19 | 800 |
| Balance Simple | 64 | 19 | 800 |
| Crawl | 51 | 19 | 700 |
| Hurdle | 51 | 19 | 700 |
| Maze | 51 | 19 | 1200 |
| Pole | 51 | 19 | 700 |
| Reach | 57 | 19 | 12000 |
| Run | 51 | 19 | 700 |
| Sit Simple | 51 | 19 | 750 |
| Sit Hard | 64 | 19 | 750 |
| Slide | 51 | 19 | 700 |
| Stair | 51 | 19 | 700 |
| Stand | 51 | 19 | 800 |
| Walk | 51 | 19 | 700 |

**ODRL.** ODRL (Lyu et al., 2024c) is a benchmark for off-dynamics RL, where the agent can interact with a source domain (with sufficient data) and a target domain (with a limited budget) at the same time, and it aims at enhancing its performance in the target domain by using source domain data. There exist dynamics discrepancies between the source domain and the target domain, therefore the source domain data is OOD for training policies in the target domain. We adopt 8 tasks from ODRL, which cover 4 kinds of dynamics shifts between the source domain and the target domain (friction/gravity/kinematic/morphology). All methods are run for 1M environment steps in the source domain, and the agent can interact with the target domain every 10 source domain steps. The action repeat is set to be 1. Following ODRL, we calculate the normalized return of the agent for better comparison, which is given by:

$$NS = \frac{J_\pi - J_r}{J_e - J_r} \times 100, \tag{64}$$

where $J_\pi$ is the episode return of the learned policy, $J_r$ is the return of the random policy, $J_e$ is the return of the expert policy. The reference random policy scores and expert policy scores for each task are outlined in Table 6. We also present in Table 7 the detailed numerical result comparison between CIR and other off-dynamics RL baselines on ODRL tasks.

Table 6: **Reference scores in ODRL tasks.**

| Task | Reference random policy score | Reference expert policy score |
|------|-------------------------------|-------------------------------|
| walker2d-friction-0.5 | 10.08 | 4229.348 |
| walker2d-friction-5.0 | 10.08 | 4988.835 |
| ant-friction-0.5 | -325.6 | 8301.338 |
| walker2d-gravity-0.5 | 10.08 | 5194.713 |
| ant-gravity-0.5 | -325.6 | 4317.065 |
| ant-gravity-5.0 | -325.6 | 6226.89 |
| ant-kinematic-anklejnt-medium | -325.6 | 5139.832 |
| ant-morph-alllegs-hard | -325.6 | 5139.832 |

Table 7: **Numerical results of CIR against off-dynamics RL baselines on ODRL tasks.** We **bold** the best mean results.

| Task | SAC | DARC | PAR | CIR (ours) |
|------|-----|------|-----|-----------|
| walker2d-friction-0.5 | 3761.6±671.2 | 4062.2±727.5 | 4074.5±225.8 | **4089.1**±628.4 |
| walker2d-friction-5.0 | 319.9±21.0 | 291.6±36.2 | **481.2**±199.6 | 319.1±67.6 |
| ant-friction-0.5 | 5664.5±274.7 | 6230.9±180.8 | 7135.0±359.5 | **7716.6**±112.3 |
| walker2d-gravity-0.5 | 2746.5±638.2 | 3112.6±367.7 | 3328.5±356.7 | **3759.8**±419.9 |
| ant-gravity-0.5 | 682.9±247.6 | 110.1±60.4 | **1737.8**±634.4 | 1203.9±627.4 |
| ant-gravity-5.0 | 4470.0±248.5 | 1312.3±236.1 | 4668.6±419.1 | **4970.3**±271.6 |
| ant-kinematic-anklejnt-medium | 6074.2±212.4 | 5535.5±393.6 | 5959.4±305.9 | **6221.3**±231.6 |
| ant-morph-alllegs-hard | 911.2±38.8 | 440.8±132.6 | 910.9±19.7 | **930.5**±20.6 |

# D HYPERPARAMETER SETUP AND EXPERIMENTAL DETAILS

## D.1 HYPERPARAMETER SETUP

We present the detailed hyperparameter setup for CIR across all tasks in Table 8.

Table 8: **Hyperparameter setup for CIR.** For most hyperparameters, we follow the hyperparameter setup used in prior works (e.g., TD3, SAC) and do not alter them.

| Hyperparameter | Value |
|---|---|
| Actor network | $(512, 512)$ |
| Actor architecture type | vanilla MLP |
| Number of hidden layers in actor | 2 |
| Critic network | $(512, 512)$ |
| Critic architecture type | U-shape |
| Number of up-sampling layers in critic | 2 |
| Number of down-sampling layers in critic | 2 |
| Batch size | 256 |
| Learning rate | $3 \times 10^{-4}$ |
| Optimizer | Adam (Kingma & Ba, 2014) |
| Discount factor | Heuristic |
| Replay buffer size | $10^6$ |
| Warmup steps | $5 \times 10^3$ |
| Nonlinearity | elu |
| Target update rate | $5 \times 10^{-3}$ |
| Entropy target | $-\dim(\mathcal{A})$ |
| Entropy auto-tuning | True |
| Maximum log std | 2 |
| Minimum log std | $-20$ |
| SMR ratio $M$ | 2 |
| Convex Q-learning coefficient $\lambda$ | 0.3 (DMC, ODRL), 1.0 (HumanoidBench) |
| AvgRNorm coefficient $c$ | 0.1 |

## D.2 EXPERIMENTAL DETAILS

In this part, we introduce the detailed experiment setup in the main text. We run CIR for 10 random seeds, and the remaining baseline results are taken directly from the SimBa paper (Lee et al., 2025a)[3]. In Section 6.3, we conduct an ablation study on the following tasks: `acrobot swingup, finger spin, finger turn easy, finger turn hard, hopper hop, hopper stand, walker run, humanoid run, humanoid stand, humanoid walk, dog run, dog walk, dog stand, dog trot`, where we include all 7 DMC hard tasks, incurring a total of 14 tasks.

In Section 6.4, we utilize the following DMC tasks: `acrobot swingup, cheetah run, fish swim, hopper hop, hopper stand, walker run, humanoid run, humanoid stand, humanoid walk` for scaling experiments.

---

[3]A full list of results can be found in https://github.com/SonyResearch/simba/tree/master/results

# E    EXTENDED EXPERIMENTS AND RESULTS

In this section, we provide missing numerical results, more ablation studies, and scaling experiments to further demonstrate the scaling ability of our proposed CIR method and verify our design choices.

## E.1    MISSING DETAILED NUMERICAL RESULTS

We first present the detailed numerical results of CIR against the following variants in Table 9:

- CIR (sigmoid): which replaces the `Tanh` function in CIR with the sigmoid function. We note that `Tanh` is equivalent to sigmoid to some extent, i.e., $\tanh(x) = \frac{e^x - e^{-x}}{e^x + e^{-x}} = \frac{2}{1+e^{-2x}} - 1 = 2sigmoid(2x) - 1$.
- CIR (softmax): which replaces the `Tanh` function in CIR with the softmax function
- CIR (layernorm): which replaces the `Tanh` function in CIR with Layer Normalization (removing $\gamma$ and $\beta$ in layer normalization for scaling).
- CIR (noSC): which removes the skip connection module in CIR.

These results correspond to the ablation study in Figure 5 (right). Intuitively, sigmoid and softmax may not be good candidates for this since they do not preserve signs. The results show that CIR (sigmoid) and CIR (softmax) exhibit poor performance on numerous tasks, which validates the rationality and advantages of using Tanh. Please note that we are not arguing that `Tanh` is the best choice; it is fully possible that there exists a better normalizer than Tanh, which we believe can be an interesting topic to study.

Table 9: **Numerical results of CIR against its variants.** SC denotes skip connection.

| Task | CIR (Tanh) | CIR (sigmoid) | CIR (softmax) | CIR (layernorm) | CIR (no SC) |
|---|---|---|---|---|---|
| acrobot-swingup | 443.5±79.0 | 132.0±82.8 | 72.5±84.5 | 251.4±22.4 | 385.1±62.9 |
| cartpole-balance | 998.1±2.4 | 996.1±2.7 | 999.0±0.8 | 999.2±0.4 | 998.9±1.1 |
| cartpole-balance-sparse | 999.6±1.2 | 1000.0±0.0 | 1000.0±0.0 | 914.6±147.9 | 812.4±280.4 |
| cartpole-swingup | 875.2±4.5 | 877.7±1.5 | 872.9±6.5 | 880.9±0.5 | 880.8±0.4 |
| cartpole-swingup-sparse | 839.6±12.3 | 750.7±72.6 | 714.0±113.2 | 833.9±14.1 | 797.7±87.1 |
| cheetah-run | 747.0±73.2 | 575.9±80.5 | 530.6±11.9 | 724.1±70.1 | 762.8±25.5 |
| finger-spin | 940.0±35.6 | 921.4±3.5 | 732.3±143.3 | 888.4±158.7 | 779.5±111.6 |
| finger-turn-easy | 892.5±88.9 | 764.8±135.4 | 172.2±120.6 | 920.4±46.9 | 862.4±111.5 |
| finger-turn-hard | 888.4±74.3 | 310.4±204.7 | 75.9±68.4 | 916.8±43.9 | 894.4±74.0 |
| fish-swim | 781.8±21.9 | 633.5±65.4 | 63.1±28.4 | 788.4±4.7 | 758.7±25.6 |
| hopper-hop | 280.1±87.9 | 90.9±6.1 | 8.9±15.9 | 250.9±88.2 | 278.1±172.8 |
| hopper-stand | 822.0±211.1 | 830.0±36.7 | 4.8±6.3 | 796.4±217.5 | 741.3±288.9 |
| pendulum-swingup | 838.0±27.9 | 488.7±400.5 | 793.5±39.6 | 814.7±42.3 | 844.5±26.1 |
| quadruped-run | 807.0±31.7 | 524.9±57.1 | 457.0±25.7 | 834.5±53.5 | 748.8±42.4 |
| quadruped-walk | 933.8±18.6 | 913.9±32.3 | 864.3±53.7 | 946.4±6.7 | 948.3±12.0 |
| reacher-easy | 974.1±29.4 | 977.3±6.6 | 954.3±50.0 | 979.3±3.5 | 987.0±1.2 |
| reacher-hard | 954.2±36.9 | 955.7±37.6 | 934.4±53.1 | 965.6±11.5 | 927.5±52.3 |
| walker-run | 810.0±14.0 | 584.6±86.4 | 566.6±54.7 | 785.5±1.3 | 784.8±16.9 |
| walker-walk | 973.9±3.1 | 967.6±4.4 | 969.0±6.2 | 972.0±2.8 | 972.2±4.2 |
| walker-stand | 986.7±5.0 | 973.1±20.5 | 982.1±3.9 | 987.6±3.7 | 967.2±12.8 |
| Average Return | 839.3 | 713.5 | 588.4 | 822.6 | 806.6 |

Furthermore, we investigate how beneficial `Tanh` would be when adopted by simpler algorithms. To that end, we run experiments on 20 DMC easy & medium tasks. We use the following baselines:

- SAC+Tanh, where we constrain the initial representations in SAC with vanilla Tanh.
- SAC+Tanh+Norm, where we add AvgRNorm and LayerNorm before constraining the initial representation while keeping the downstream network architecture unchanged.

We summarize the results in Table 10. We find that SAC+Tanh incurs a performance drop on many tasks, despite the fact that `Tanh` still improves its average performance against vanilla SAC, indicating that `Tanh` alone can still be beneficial to simple algorithms. By adding normalization

methods, the performance of SAC+Tanh improves drastically and significantly outperforms vanilla SAC, verifying our Theorem 4.5 and showing that many components (`Tanh`, normalization, etc.) work altogether to enable CIR to perform well. Furthermore, SAC+Tanh+Norm underperforms CIR on numerous tasks, which validates our design choices.

Table 10: **Numerical results of CIR against SAC variants.**

| Task | SAC | SAC+Tanh | SAC+Tanh+Norm | CIR |
|---|---|---|---|---|
| acrobot-swingup | 57.6 | 18.2±17.2 | 5.5±4.3 | 443.5±79.0 |
| cartpole-balance | 998.8 | 999.8±0.1 | 999.8±0.0 | 998.1±2.4 |
| cartpole-balance-sparse | 1000.0 | 1000.0±0.0 | 1000.0±0.0 | 999.6±1.2 |
| cartpole-swingup | 863.2 | 880.4±1.2 | 876.7±7.2 | 875.2±4.5 |
| cartpole-swingup-sparse | 780.0 | 650.4±326.0 | 788.9±32.9 | 839.6±12.3 |
| cheetah-run | 716.4 | 642.2±69.5 | 749.1±63.1 | 747.0±73.2 |
| finger-spin | 814.7 | 832.7±44.7 | 851.7±21.2 | 940.0±35.6 |
| finger-turn-easy | 903.1 | 765.4±197.7 | 791.6±109.6 | 892.5±88.9 |
| finger-turn-hard | 775.2 | 798.4±155.1 | 619.9±244.9 | 888.4±74.3 |
| fish-swim | 462.7 | 730.5±37.6 | 639.5±116.3 | 781.8±21.9 |
| hopper-hop | 159.4 | 52.4±44.6 | 136.5±89.3 | 280.1±87.9 |
| hopper-stand | 845.9 | 468.4±256.3 | 528.0±334.2 | 822.0±211.1 |
| pendulum-swingup | 476.6 | 676.0±293.3 | 833.3±29.1 | 838.0±27.9 |
| quadruped-run | 116.9 | 565.6±114.1 | 837.8±52.3 | 807.0±31.7 |
| quadruped-walk | 147.8 | 777.8±296.4 | 938.1±20.5 | 933.8±18.6 |
| reacher-easy | 951.8 | 981.3±3.9 | 981.4±2.4 | 974.1±29.4 |
| reacher-hard | 959.6 | 912.4±54.0 | 968.8±2.2 | 954.2±36.9 |
| walker-run | 629.4 | 638.7±56.7 | 683.2±36.5 | 810.0±14.0 |
| walker-walk | 956.7 | 961.2±7.1 | 938.0±55.4 | 973.9±3.1 |
| walker-stand | 972.6 | 972.1±1.5 | 981.0±6.8 | 986.7±5.0 |

Furthermore, we replace the UTD replay method in SimBa with SMR in the official SimBa codebase, giving birth to SimBa (SMR). This aims at understanding the importance of SMR and making a fair comparison between CIR and SimBa. We run SimBa (SMR) on 20 DMC suite easy and medium tasks. We compare CIR (SMR) against SimBa (UTD) and SimBa (SMR) below in Table 11. We run SimBa (SMR) for 5 seeds. The results show that SMR can boost the performance of SimBa on some tasks, while it can also incur performance degradation on many tasks compared to UTD. It seems that SMR can work better by combining it with CIR. The reasons can lie in different network architectures and algorithmic components between CIR and SimBa.

### E.2 MORE ABLATION STUDY

Following the main text, we adopt 7 DMC medium tasks and 7 DMC hard tasks (specified in Appendix D.2) and consider the following CIR variants: (i) **ActorLN**, which incorporates layer normalization to the actor network; (ii) **ActorCIR**, which adopts the same network architecture for the actor as the critics in CIR; (iii) **Res**, which substitutes skip connections for residual connections; (iv) **NoEnt**, which removes the entropy term in Equation 10. We summarize the overall results in Figure 7, where we report the percent CIR performance (% CIR performance) of these variants. If the percent CIR performance exceeds 100, it indicates that the variant outperforms CIR, and underperforms CIR if it is smaller than 100.

**On the actor network architecture.** We find that both **ActorLN** and **ActorCIR** underperform CIR, indicating that either adding layer normalization to the actor network or replacing the plain MLP network with the critics' network architecture in CIR is not effective. A similar phenomenon is observed in BRO (Nauman et al., 2024). The results undoubtedly support our design choice in Section 5.2.

**On the critic network architecture.** We find that **Res** achieves similar performance as CIR on DMC hard tasks but exhibits inferior performance on medium tasks. This verifies our design choice on the U-shape network.

Table 11: **Comparison of CIR against SimBa with SMR.**

| Task | CIR (SMR) | SimBa (UTD) | SimBa (SMR) |
|------|-----------|-------------|-------------|
| acrobot-swingup | 443.5±79.0 | 331.6 | 212.4±78.1 |
| cartpole-balance | 998.1±2.4 | 999.1 | 997.5±1.9 |
| cartpole-balance-sparse | 999.6±1.2 | 940.5 | 993.3±11.6 |
| cartpole-swingup | 875.2±4.5 | 866.5 | 872.2±6.1 |
| cartpole-swingup-sparse | 839.6±12.3 | 824.0 | 820.0±14.6 |
| cheetah-run | 747.0±73.2 | 815.0 | 716.2±17.4 |
| finger-spin | 940.0±35.6 | 778.8 | 741.4±121.5 |
| finger-turn-easy | 892.5±88.9 | 881.3 | 874.8±13.1 |
| finger-turn-hard | 888.4±74.3 | 860.2 | 885.8±30.6 |
| fish-swim | 781.8±21.9 | 786.7 | 268.5±58.4 |
| hopper-hop | 280.1±87.9 | 326.7 | 309.1±72.4 |
| hopper-stand | 822.0±211.1 | 811.8 | 737.4±191.4 |
| pendulum-swingup | 838.0±27.9 | 824.5 | 609.9±354.5 |
| quadruped-run | 807.0±31.7 | 883.7 | 800.7±39.8 |
| quadruped-walk | 933.8±18.6 | 953.0 | 954.0±8.7 |
| reacher-easy | 974.1±29.4 | 972.2 | 711.8±78.5 |
| reacher-hard | 954.2±36.9 | 966.0 | 803.7±142.7 |
| walker-run | 810.0±14.0 | 687.2 | 702.0±14.2 |
| walker-walk | 973.9±3.1 | 970.7 | 972.5±3.2 |
| walker-stand | 986.7±5.0 | 983.0 | 977.4±0.9 |

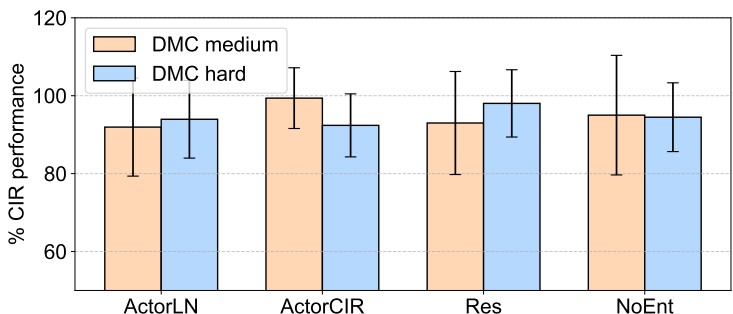

Figure 7: **Extended ablation study.** We report % CIR performance of each variant and aggregate their performance across all tasks. The results are averaged across five seeds.

**On algorithmic components.** The results show that excluding the entropy term in the target value has a minor influence on the performance of CIR. It is recommended to keep the entropy term in the target value.

We consider the following variants of CIR to further verify our design choices, (i) **NoInputLN**, where there is no layer normalization module at the initial layer, i.e., Equation 7 becomes $z_t = \tanh(\text{AvgRNorm}(\text{Linear}(o_t)))$; (ii) **NoAvgRNorm**, which removes the AvgRNorm module in CIR, i.e., Equation 7 becomes $z_t = \tanh(\text{LayerNorm}(\text{Linear}(o_t)))$; (iii) **NoInputNorm**, where we remove both AvgRNorm and layer normalization in the initial layer, i.e., Equation 7 becomes $z_t = \tanh(\text{Linear}(o_t))$; (iv) **MaxRNorm**, where we replace the AvgRNorm module in the initial layer with the maximum representation normalization, $\text{MaxRNorm}(\mathbf{x}) = \dfrac{\mathbf{x}}{\text{Max}(\tilde{\mathbf{x}})}, \tilde{\mathbf{x}} = |\mathbf{x}|$, where $\text{Max}(\cdot)$ is the maximum operator; (v) **AllAvgRNorm**, which adds the AvgRNorm module after the layer normalization on all layers; (vi) **Orthoini**, which uses orthogonal initialization for the critic network; (vii) **WD**, which adopts weight decay for the optimizer of the critic network; (viii) **AvgQ**, where we use the average Q learning for calculating the target value ($\lambda = 0.5$ in Equation 10).

To comprehensively compare the performance of CIR against these variants, we run experiments on 9 DMC easy & medium tasks, and 3 DMC hard tasks. This amounts to a total of 12 tasks. The

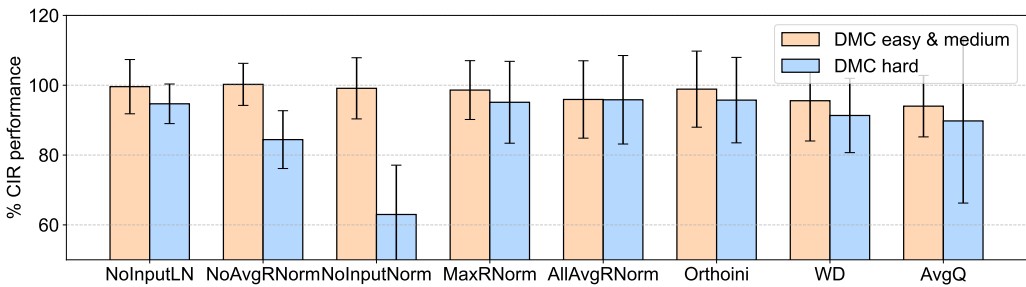

Figure 8: **Extended ablation study.** Following the main text, we report % CIR performance of each variant and aggregate their performance across all tasks. We use five random seeds for each variant.

full list of the selected tasks gives: `acrobot swingup, cheetah run, finger spin, finger turn easy, finger turn hard, fish swim, hopper hop, hopper stand, walker run, humanoid run, humanoid stand, humanoid walk`.

Following the main text, we also report % CIR performance of these variants at the final environment step. The empirical results can be found in Figure 8.

It turns out that both layer normalization and the AvgRNorm modules are critical to CIR, especially on complex tasks like humanoid. Adopting MaxRNorm or orthogonal initialization leads to a similar performance as the vanilla CIR on DMC easy & medium tasks, but slightly inferior performance on hard tasks. **WD** and **AvgQ**, however, can impede the performance improvement of the agent. The above evidence clearly verifies our design choices in the CIR framework.

Furthermore, we are interested in investigating how sensitive CIR is to the choices of the AvgRNorm parameter $c$. In CIR, we set $c = 0.1$ by default. Intuitively, using a larger $c$ is not preferred both theoretically and empirically, since our theoretical analysis often requires a small $c$, and using a large $c$ can make the resulting representation element lie in the gradient vanishing region of the `Tanh` function, and harm the performance of the agent eventually. We then conduct experiments on DMC tasks with varying $c$, $c \in \{0.01, 0.1, 0.2, 0.5, 1.0\}$.

To be specific, we run CIR with different choices of $c$ on the following tasks: `acrobot swingup, cheetah run, fish swim, hopper hop, hopper stand, walker run, dog run, dog stand, dog walk, humanoid run, humanoid stand, humanoid walk`. These tasks include 6 DMC easy & medium tasks and 6 DMC hard tasks, which we believe can reveal how sensitive CIR is to the choices of $c$ under varying task complexities. We summarize the empirical results in Figure 9 where we report the average episode return in conjunction with the aggregated standard deviations (averaged across all environments). The results on DMC easy & medium tasks show that CIR is comparatively insensitive to $c$ on those tasks, while the results in DMC hard tasks show that using a large $c$ can incur a significant performance drop, verifying our claims above. We then use $c = 0.1$ by default across all tasks.

### E.3 MORE SCALING RESULTS

In this part, we first investigate the performance of CIR under distinct critic network depth by using fewer or more up-sampling layers and down-sampling layers in the U-shape network. Denote the number of up-sampling layers as $L$, we compare the performance of CIR under $L \in \{1, 2, 3\}$, where $L = 1$ indicates that there is only 1 up-sampling layer and 1 down-sampling layer, while $L = 3$ means that there are 3 up-sampling and down-sampling layers, respectively (CIR adopts $L = 2$ by default). We use 6 DMC easy & medium tasks, and 7 DMC hard tasks for experiments. The complete task list includes: `acrobot swingup, cheetah run, fish swim, hopper hop, hopper stand, walker run, dog run, dog trot, dog stand, dog walk, humanoid run, humanoid stand, humanoid walk`.

We report the aggregated average performance along with the aggregated standard deviations of CIR under different equivalent environment steps in Figure 10 (left). The results show that using fewer

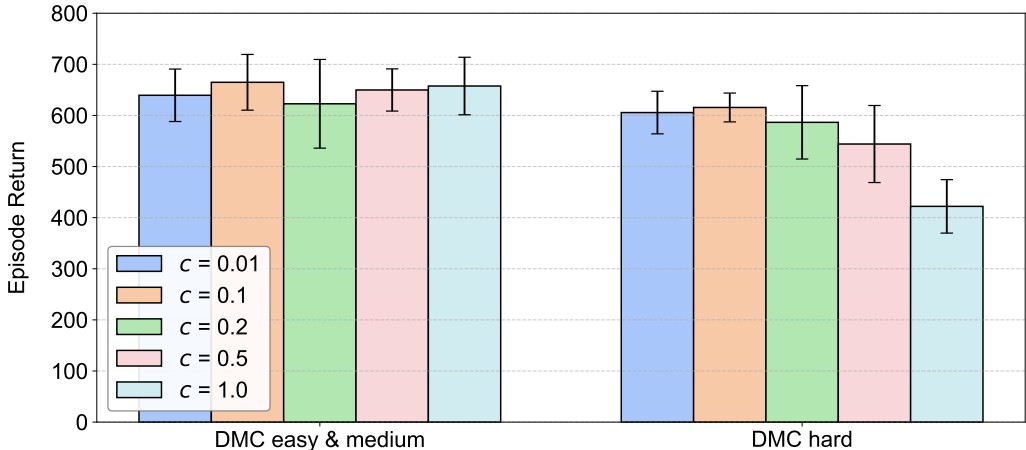

Figure 9: **Parameter study on AvgRNorm.** We conduct experiments on numerous DMC tasks to investigate the influence of the introduced hyperparameter $c$. We report the aggregated average performance along with the aggregated standard deviations. We use five random seeds.

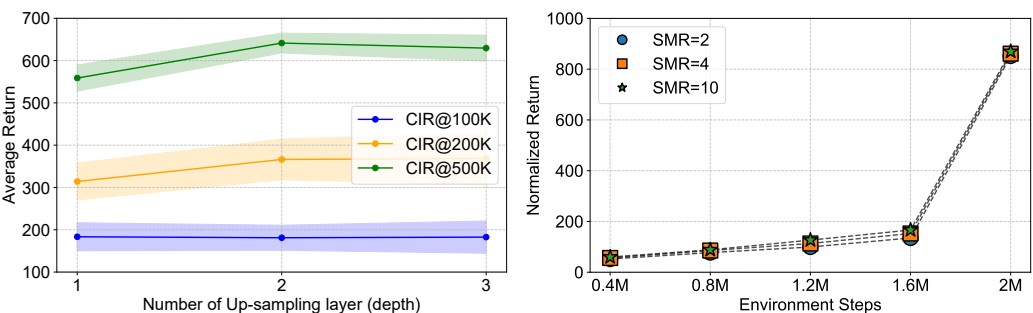

Figure 10: **Extended scaling results of CIR.** We include the scaling results on network depth (**left**) and the SMR ratio (**right**). We report the aggregated average return results for DMC tasks and aggregated normalized return results for HumanoidBench tasks. All results are obtained by averaging across five different seeds and the shaded region captures the standard deviation.

up-sampling layers can incur a performance drop while increasing the number of up-sampling layers from 2 to 3 almost brings no performance improvement. We hence use $L = 2$ by default.

We also include SMR ratio scaling experiments on some HumanoidBench tasks, `h1-reach-v0`, `h1-run-v0`, `h1-walk-v0`, `h1-slide-v0`. We run CIR on these tasks with the SMR ratio $M \in \{2, 4, 10\}$ for 2M environment steps. The aggregated normalized return results in Figure 10 (right) show that CIR is somewhat insensitive to $M$ on the selected HumanoidBench tasks, despite the fact that using a larger SMR ratio $M$ can still bring performance improvements. Notably, CIR scales the replay ratio without any need for parameter resetting tricks.

### E.4 NUMERICAL RESULTS FOR ABLATION STUDIES

In this part, we list all missing detailed numerical results in our ablation study for a better understanding of the learning dynamics of all CIR variants. We present in Table 12 the detailed results of CIR variants on 7 DMC medium tasks and 7 DMC hard tasks, which correspond to the results presented in Figure 5 (left). In Table 13, we show the numerical results of the ablation study in Figure 7. In Table 14, we include the numerical results that correspond to Figure 8 in Appendix E.2.

Table 12: **Numerical results of CIR and its variants on 7 DMC medium tasks and 7 DMC hard tasks in Figure 5.** We report the average return and the standard deviations.

| Task | NoTanh | NoLN | CDQ | UTD | CIR |
|---|---|---|---|---|---|
| acrobot-swingup | 461.7±49.8 | 458.1±81.9 | 9.1±7.3 | 494.4±22.1 | 443.5±79.0 |
| finger-spin | 879.4±119.4 | 828.4±201.6 | 924.0±36.9 | 967.9±30.4 | 940.0±35.6 |
| finger-turn-easy | 910.0±60.3 | 970.1±6.2 | 897.7±80.0 | 880.9±90.3 | 892.5±88.9 |
| finger-turn-hard | 914.6±54.3 | 943.6±42.8 | 809.5±141.5 | 895.7±70.2 | 888.4±74.3 |
| hopper-hop | 171.0±95.6 | 140.2±111.3 | 35.1±30.8 | 250.1±54.7 | 280.1±87.9 |
| hopper-stand | 749.6±208.1 | 709.2±257.0 | 448.4±329.7 | 760.4±158.8 | 822.0±211.1 |
| walker-run | 821.0±7.4 | 798.8±16.6 | 792.8±19.9 | 795.2±5.5 | 810.0±14.0 |
| dog-run | 259.0±26.3 | 18.1±17.9 | 386.6±48.6 | 299.4±33.3 | 326.8±30.6 |
| dog-stand | 796.0±33.7 | 956.0±4.8 | 919.4±44.1 | 855.4±40.3 | 937.3±23.8 |
| dog-trot | 392.6±68.9 | 219.9±44.2 | 687.5±124.2 | 419.4±133.2 | 685.1±100.0 |
| dog-walk | 645.5±52.8 | 618.5±97.5 | 909.7±22.3 | 763.0±188.9 | 880.5±38.7 |
| humanoid-run | 175.7±19.9 | 129.7±14.8 | 1.4±0.6 | 162.7±31.5 | 174.2±22.4 |
| humanoid-stand | 838.4±37.9 | 761.8±103.0 | 583.7±337.1 | 785.9±84.8 | 846.3±35.4 |
| humanoid-walk | 618.0±64.4 | 536.7±22.8 | 550.2±53.0 | 520.3±39.3 | 528.5±18.2 |

Table 13: **Numerical results of CIR and its variants on 7 DMC medium tasks and 7 DMC hard tasks in Figure 7.** We report the average return and the standard deviations.

| Task | ActorLN | ActorCIR | Res | NoEnt | CIR |
|---|---|---|---|---|---|
| acrobot-swingup | 313.4±115.6 | 459.7±20.4 | 423.8±110.3 | 512.2±117.4 | 443.5±79.0 |
| finger-spin | 948.7±22.3 | 984.6±4.5 | 914.1±122.5 | 925.7±97.2 | 940.0±35.6 |
| finger-turn-easy | 902.8±82.7 | 899.4±100.5 | 929.6±48.7 | 816.1±84.4 | 892.5±88.9 |
| finger-turn-hard | 870.0±55.2 | 793.3±137.9 | 851.1±79.1 | 885.8±63.9 | 888.4±74.3 |
| hopper-hop | 279.9±113.8 | 189.8±105.0 | 62.2±62.5 | 258.2±131.7 | 280.1±87.9 |
| hopper-stand | 538.8±241.9 | 910.3±17.5 | 765.5±232.1 | 612.2±278.4 | 822.0±211.1 |
| walker-run | 813.9±7.4 | 808.1±10.0 | 774.8±16.3 | 813.0±6.3 | 810.0±14.0 |
| dog-run | 346.1±39.9 | 260.3±39.3 | 325.9±74.7 | 305.9±31.2 | 326.8±30.6 |
| dog-stand | 946.6±37.8 | 921.5±54.3 | 933.8±48.5 | 961.6±10.7 | 937.3±23.8 |
| dog-trot | 603.4±128.6 | 486.9±83.4 | 666.7±56.7 | 561.2±191.4 | 685.1±100.0 |
| dog-walk | 884.4±22.5 | 909.9±23.6 | 894.7±28.5 | 882.5±37.7 | 880.5±38.7 |
| humanoid-run | 148.2±11.3 | 177.7±27.9 | 162.5±18.1 | 135.8±7.5 | 174.2±22.4 |
| humanoid-stand | 592.1±87.9 | 811.6±37.7 | 770.4±103.9 | 799.7±61.5 | 846.3±35.4 |
| humanoid-walk | 593.1±108.5 | 477.4±88.2 | 538.3±47.9 | 490.4±46.5 | 528.5±18.2 |

Table 14: **Numerical results of CIR and its variants on 9 DMC medium tasks and 3 DMC hard tasks in Figure 8.** We report the average return and the standard deviations.

| Task | NoInputLN | NoAvgRNorm | NoInputNorm | MaxRNorm | AllAvgRNorm | Orthoini | WD | AvgQ | CIR |
|---|---|---|---|---|---|---|---|---|---|
| acrobot-swingup | 528.8±44.6 | 475.2±47.4 | 475.3±68.0 | 473.0±70.4 | 473.7±94.9 | 437.4±55.8 | 410.4±74.6 | 475.9±20.6 | 443.5±79.0 |
| cheetah-run | 704.0±23.6 | 722.5±40.5 | 713.7±70.4 | 652.4±117.8 | 717.4±19.7 | 683.1±135.1 | 714.9±37.2 | 753.6±68.8 | 747.0±73.2 |
| finger-spin | 867.8±125.8 | 970.3±15.3 | 954.4±16.1 | 975.6±14.8 | 962.3±23.6 | 967.9±23.2 | 974.7±16.2 | 969.0±27.9 | 940.0±35.6 |
| finger-turn-easy | 898.0±44.4 | 944.9±36.8 | 916.4±42.6 | 890.5±78.1 | 895.8±45.8 | 873.9±98.6 | 896.1±82.3 | 892.7±37.7 | 892.5±88.9 |
| finger-turn-hard | 870.1±71.0 | 916.6±53.1 | 930.2±54.2 | 872.4±72.6 | 838.2±137.0 | 827.4±144.7 | 874.6±73.0 | 769.2±146.2 | 888.4±74.3 |
| fish-swim | 778.0±23.3 | 742.4±25.7 | 784.5±19.3 | 758.0±20.6 | 733.3±43.0 | 768.5±29.2 | 739.6±44.4 | 768.5±29.2 | 781.8±21.9 |
| hopper-hop | 249.9±125.6 | 224.8±76.3 | 258.5±70.0 | 202.4±86.7 | 183.7±35.3 | 274.2±152.1 | 259.9±93.3 | 109.8±62.7 | 280.1±87.9 |
| hopper-stand | 892.0±40.4 | 869.4±58.0 | 775.0±193.9 | 883.9±77.5 | 730.8±325.3 | 893.1±52.8 | 638.2±326.8 | 653.1±175.2 | 822.0±211.1 |
| walker-run | 789.8±14.6 | 755.7±44.9 | 739.0±44.1 | 805.6±17.9 | 801.0±7.3 | 805.7±28.2 | 803.8±13.9 | 817.8±13.4 | 810.0±14.0 |
| humanoid-run | 159.8±12.1 | 172.3±5.5 | 103.9±10.1 | 182.5±34.1 | 139.0±19.2 | 170.6±25.4 | 129.5±62.6 | 167.1±47.4 | 174.2±22.4 |
| humanoid-stand | 815.3±41.0 | 684.0±94.0 | 521.2±167.2 | 767.5±124.0 | 819.0±34.4 | 770.7±105.8 | 758.8±79.2 | 805.8±66.0 | 846.3±35.4 |
| humanoid-walk | 491.5±34.7 | 451.5±28.7 | 350.4±41.5 | 523.3±23.5 | 526.4±142.7 | 541.7±58.3 | 526.4±23.0 | 417.7±251.0 | 528.5±18.2 |

# F LEARNING CURVES

The full learning curves of CIR on DMC easy & medium tasks can be found in Figure 11, the learning curves on DMC hard tasks are shown in Figure 12, the learning curves on HumanoidBench tasks are available in Figure 13, and the learning curves on ODRL tasks are depicted in Figure 14. We also include learning curves of other baseline methods on all tasks.

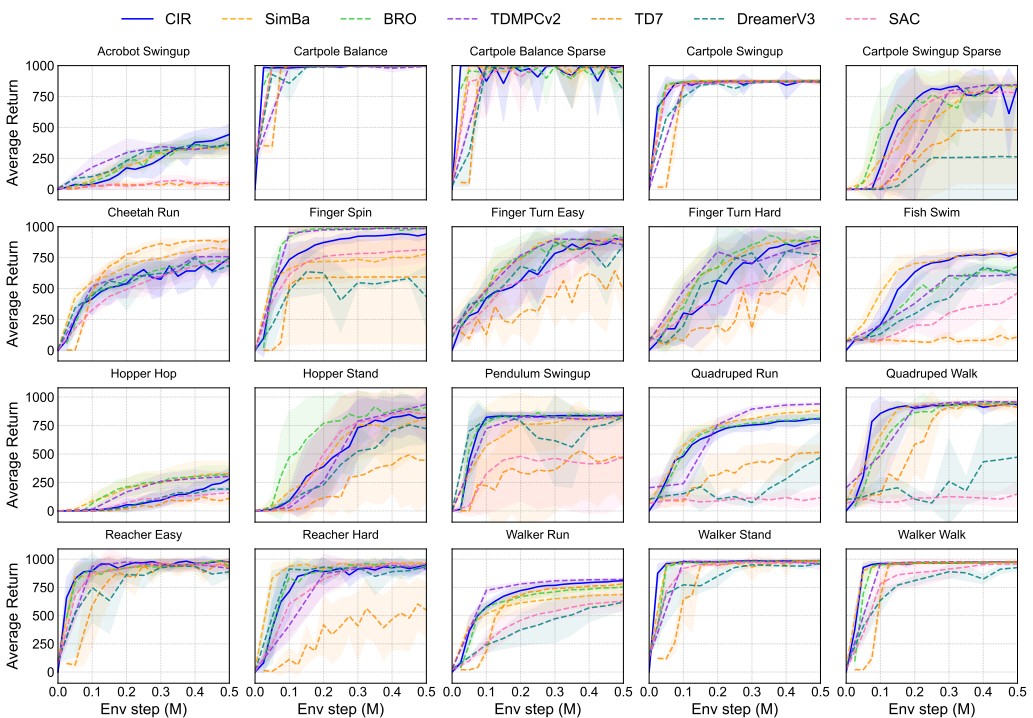

Figure 11: **Learning curves of CIR against baseline methods on DMC easy & medium tasks.** The shaded region captures the standard deviation. We run CIR for 10 random seeds.

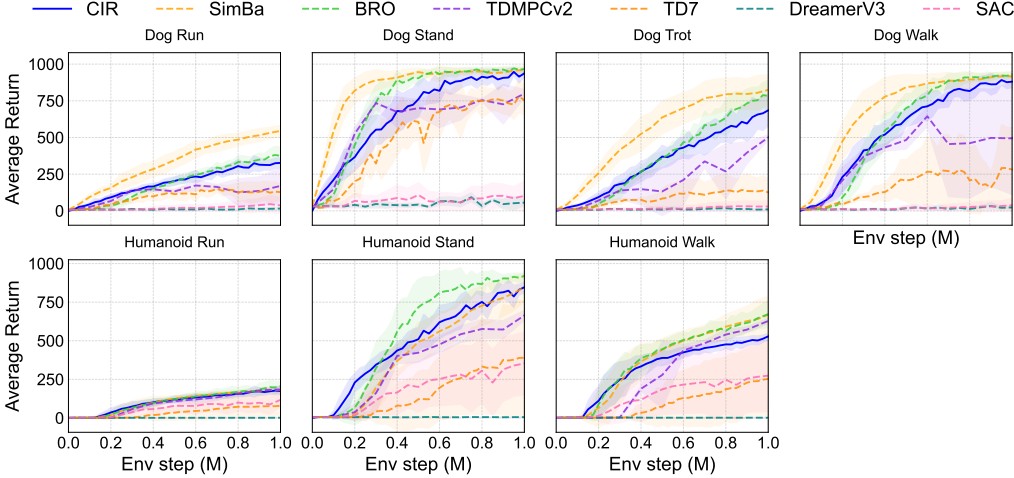

Figure 12: **Learning curves of CIR against baseline methods on DMC hard tasks.** The shaded region denotes the standard deviation. The results of CIR are averaged across 10 different random seeds.

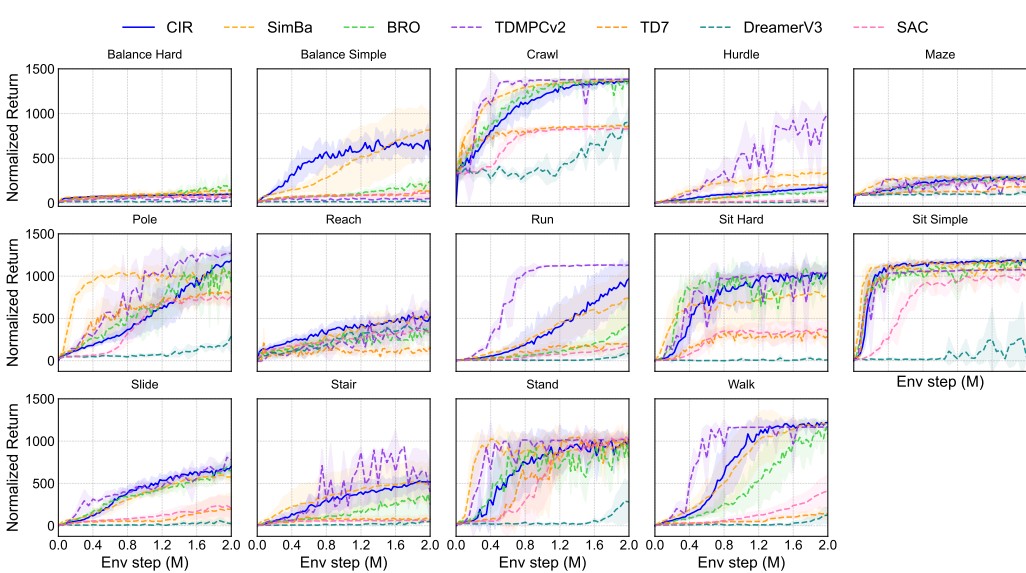

Figure 13: **Learning curves of CIR against other baseline methods on HumanoidBench tasks.** The shaded region represents the standard deviations. The results of CIR are acquired by averaging across 10 random seeds.

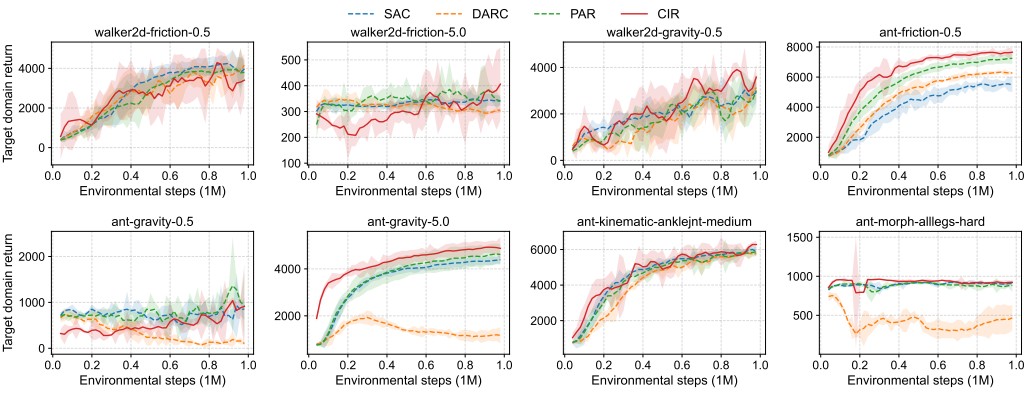

Figure 14: **Learning curves of CIR against other baseline methods on ODRL tasks.** The shaded region represents the standard deviations.

## G    FULL RESULTS

In this part, we include the detailed numerical results of CIR against baseline methods on each DMC task and HumanoidBench task. We report the average episode return on DMC easy & medium/hard tasks in Table 15 and Table 16, respectively, and normalized return results on HumanoidBench tasks in Table 17. Apart from the average return across all evaluated tasks, we also report statistical metrics like IQM (interquartile mean), median, and OG (optimality gap) (Agarwal et al., 2021). The results show that CIR can achieve competitive results compared to strong baselines like SimBa, TD-MPC2, and can even outperform them on some tasks.

Table 15: **Detailed results for DeepMind Control Suite Easy & Medium Tasks**. Results for CIR are averaged over 10 seeds and other results are taken directly from the SimBa paper (Lee et al., 2025a).

| Task | CIR | SimBa | BRO | TD7 | SAC | TD-MPC2 | DreamerV3 |
|---|---|---|---|---|---|---|---|
| Acrobot Swingup | 443.53 | 331.57 | 390.78 | 39.83 | 57.56 | 361.07 | 360.46 |
| Cartpole Balance | 998.05 | 999.05 | 998.66 | 998.79 | 998.79 | 993.93 | 994.95 |
| Cartpole Balance Sparse | 999.60 | 940.53 | 954.21 | 988.58 | 1000.00 | 1000.00 | 800.25 |
| Cartpole Swingup | 875.22 | 866.53 | 878.69 | 878.09 | 863.24 | 876.07 | 863.90 |
| Cartpole Swingup Sparse | 839.56 | 823.97 | 833.18 | 480.74 | 779.99 | 844.77 | 262.69 |
| Cheetah Run | 747.00 | 814.97 | 739.67 | 897.76 | 716.43 | 757.60 | 686.25 |
| Finger Spin | 940.05 | 778.83 | 987.45 | 592.74 | 814.69 | 984.63 | 434.06 |
| Finger Turn Easy | 892.50 | 881.33 | 905.85 | 485.66 | 903.07 | 854.67 | 851.11 |
| Finger Turn Hard | 888.40 | 860.22 | 905.30 | 596.32 | 775.21 | 876.27 | 769.86 |
| Fish Swim | 781.79 | 786.73 | 680.34 | 108.84 | 462.67 | 610.23 | 603.34 |
| Hopper Hop | 280.10 | 326.69 | 315.04 | 110.27 | 159.43 | 303.27 | 192.70 |
| Hopper Stand | 821.98 | 811.75 | 910.88 | 445.50 | 845.89 | 936.47 | 722.42 |
| Pendulum Swingup | 837.95 | 824.53 | 816.20 | 461.40 | 476.58 | 841.70 | 825.17 |
| Quadruped Run | 807.00 | 883.68 | 818.62 | 515.13 | 116.91 | 939.63 | 471.25 |
| Quadruped Walk | 933.80 | 952.96 | 936.06 | 910.55 | 147.83 | 957.17 | 472.31 |
| Reacher Easy | 974.06 | 972.23 | 933.77 | 920.38 | 951.80 | 919.43 | 888.36 |
| Reacher Hard | 954.21 | 965.96 | 956.52 | 549.50 | 959.59 | 913.73 | 935.25 |
| Walker Run | 810.04 | 687.16 | 754.43 | 782.32 | 629.44 | 820.40 | 620.13 |
| Walker Stand | 986.66 | 983.02 | 986.58 | 984.63 | 972.59 | 957.17 | 963.28 |
| Walker Walk | 973.86 | 970.73 | 973.41 | 976.58 | 956.67 | 978.70 | 925.46 |
| IQM | 887.86 | 885.70 | 888.02 | 740.19 | 799.75 | 895.47 | 748.58 |
| Median | 840.04 | 816.78 | 836.62 | 623.18 | 676.59 | 836.93 | 684.18 |
| Mean | 839.27 | 823.12 | 833.78 | 636.18 | 679.42 | 836.34 | 682.16 |
| OG | 0.1607 | 0.1769 | 0.1662 | 0.3638 | 0.3206 | 0.1637 | 0.3178 |

Table 16: **Detailed results for DeepMind Control Suite Hard Tasks**. Results for CIR are averaged over 10 seeds and other results are taken directly from the SimBa paper (Lee et al., 2025a).

| Task | CIR | SimBa | BRO | TD7 | SAC | TD-MPC2 | DreamerV3 |
|---|---|---|---|---|---|---|---|
| Dog Run | 326.80 | 544.86 | 374.63 | 127.48 | 36.86 | 169.87 | 15.72 |
| Dog Stand | 937.34 | 960.38 | 966.97 | 753.23 | 102.04 | 798.93 | 55.87 |
| Dog Trot | 685.11 | 824.69 | 783.12 | 126.00 | 29.36 | 500.03 | 10.19 |
| Dog Walk | 880.48 | 916.80 | 931.46 | 280.87 | 38.14 | 493.93 | 23.36 |
| Humanoid Run | 174.25 | 181.57 | 204.96 | 79.32 | 116.97 | 184.57 | 0.91 |
| Humanoid Stand | 846.25 | 846.11 | 920.11 | 389.80 | 352.72 | 663.73 | 5.12 |
| Humanoid Walk | 528.45 | 668.48 | 672.55 | 252.72 | 273.67 | 628.23 | 1.33 |
| IQM | 669.59 | 773.28 | 771.50 | 216.04 | 69.03 | 527.11 | 9.63 |
| Median | 625.85 | 706.39 | 694.20 | 272.62 | 159.36 | 528.26 | 17.13 |
| Mean | 625.52 | 706.13 | 693.40 | 287.06 | 135.68 | 491.33 | 16.07 |
| OG | 0.3745 | 0.2939 | 0.3066 | 0.7129 | 0.8643 | 0.5087 | 0.9839 |

Table 17: **Detailed results for HumanoidBench**. Results for CIR are averaged over 10 seeds and other results are taken directly from the SimBa paper (Lee et al., 2025a).

| Task | CIR | SimBa | BRO | TD7 | SAC | TD-MPC2 | DreamerV3 |
|------|-----|-------|-----|-----|-----|---------|-----------|
| Balance Hard | 98.51 | 137.20 | 145.95 | 79.90 | 69.02 | 64.56 | 16.07 |
| Balance Simple | 597.01 | 816.38 | 246.57 | 132.82 | 113.38 | 50.69 | 14.09 |
| Crawl | 1364.81 | 1370.51 | 1373.83 | 868.63 | 830.56 | 1384.40 | 906.66 |
| Hurdle | 183.83 | 340.60 | 128.60 | 200.28 | 31.89 | 1000.12 | 18.78 |
| Maze | 286.64 | 283.58 | 259.03 | 179.23 | 254.38 | 198.65 | 114.90 |
| Pole | 1184.66 | 1036.70 | 915.89 | 830.72 | 760.78 | 1269.57 | 289.18 |
| Reach | 515.11 | 523.10 | 317.99 | 159.37 | 347.92 | 505.61 | 341.62 |
| Run | 971.70 | 741.16 | 429.74 | 196.85 | 168.25 | 1130.94 | 85.93 |
| Sit Hard | 1042.88 | 783.95 | 989.07 | 293.96 | 345.65 | 1027.47 | 9.95 |
| Sit Simple | 1197.34 | 1059.73 | 1151.84 | 1183.26 | 994.58 | 1074.96 | 40.53 |
| Slide | 701.69 | 577.87 | 653.17 | 197.07 | 208.46 | 780.82 | 24.43 |
| Stair | 517.94 | 527.49 | 249.86 | 77.19 | 65.53 | 398.21 | 49.04 |
| Stand | 1013.30 | 906.94 | 780.52 | 1005.54 | 1029.78 | 1020.59 | 280.99 |
| Walk | 1218.76 | 1202.91 | 1080.63 | 143.86 | 412.21 | 1165.42 | 125.59 |
| IQM | 809.58 | 747.43 | 558.03 | 256.77 | 311.46 | 885.67 | 72.30 |
| Median | 779.27 | 733.09 | 632.93 | 385.41 | 399.93 | 796.18 | 160.49 |
| Mean | 778.15 | 736.30 | 623.05 | 396.33 | 402.31 | 787.64 | 165.55 |
| OG | 0.3047 | 0.3197 | 0.4345 | 0.6196 | 0.6016 | 0.2904 | 0.8352 |

## H  COMPUTE INFRASTRUCTURE

In Table 18, we list the compute infrastructure that we use to run all of the algorithms.

Table 18: **Compute infrastructure.**

| CPU | GPU | Memory |
|-----|-----|--------|
| AMD EPYC 7452 | RTX3090×8 | 192GB |

## I  LIMITATIONS AND BROADER IMPACTS

**Limitations.** The limitations of CIR lie in three aspects: (i) CIR relies on U-shape critic networks, making it less convenient to modify the network depth compared to residual blocks; (ii) the performance of CIR is inferior to baselines on DMC hard tasks. However, CIR provides a distinct way compared to prior methods to achieve strong sample efficiency on numerous tasks with fixed algorithmic configuration and mostly fixed hyperparameters; (iii) we only evaluate CIR on several state-based online continuous control RL tasks, while the applicability of CIR on offline RL tasks or pixel-based RL tasks remains to be explored.

**Broader Impacts.** In this paper, we introduce a novel method called CIR to enhance the sample efficiency of off-policy RL agents. CIR successfully improves the performance of the agent by constraining its initial representations. CIR can bring new insights to the community on developing stronger and more advanced off-policy RL algorithms, and demonstrates practical potential for accelerating progress in embodied AI systems through more sample-efficient training paradigms. This work adheres to ethical AI development standards, and we have not identified significant negative societal impacts requiring special emphasis in this context.

## J  LARGE LANGUAGE MODEL (LLM) USAGE STATEMENT

No large language models were utilized in any phase of this paper's creation. This encompasses all aspects such as data analysis, literature review, language polishing, etc.

