# OpenReview forum: "Temporal Difference Learning with Constrained Initial Representations"
_ICLR.cc/2026/Conference — ICLR 2026 Conference Withdrawn Submission_

### Official Review · Reviewer_huEK · 2025-10-22

**Soundness:** 2
**Presentation:** 2
**Contribution:** 2
**Rating:** 4
**Confidence:** 4

**Summary:**

This paper introduces new training scheme called Constrained Initial Representations (CIR) whose goal is to stabilise the TD-learning and reduce distributional shift of off-policy training. They achieve it with the help of Tanh activation function introduced in the early layers of the Q-network, which, as they justify theoretically, allows us to upper bound the variance of the semi-gradient of the TD-loss. In addition to that, they introduce two other modifications to previous works: U-net type of architecture after initial layers with Tanh, and convex-Q learning, which is presented as an alternative to clipped double Q loss. The paper presents a vast set of ablation studies for each of the algorithmic choices and demonstrates an improvement over baselines on HumanoidBench, while on DMC tasks it shows a small degradation.

While CIR indeed improves on some difficult benchmarks over baselines, not all comparisons are conclusive and in particular the choice of Tanh over LayerNorm and the need of the U-net like architecture (see Weaknesses). For all these reasons, I recommend a weak reject.

**Strengths:**

Overall, the paper is quite easy to follow and it suggests a novel neural network architecture for Q-learning based on combination of Tanh for early layers + U-net for the main body of the neural network, and they propose a novel TD loss based on convex combination between two types of aggregation of target Q-networks. To confirm the algorithmic design choices, the authors perform massive ablation studies, that confirm the necessity of regularisation layer at the beginning of the neural network and the advantage of convex Q-learning over clipped double Q-learning. The ablated parameters, while show inferior performance for some tasks with respect to baselines, manage to improve over baselines on many hard tasks.

**Weaknesses:**

As mentioned before, while CIR achieves a clear improvement on HumanoidBench tasks, in general it is less good than baselines on DMC tasks as shown in Figure 4.

The most of theoretical results are not that original considering that there are prior works that investigate the stability of TD-learning with normalisation layers. Compare to https://arxiv.org/pdf/2407.04811 (a missing literature), where they did a more nuanced analysis of TD-stabilty, where they have decoupled the instability coming from off-policy learning and non-linear function approximation, and they showed that the usage of layer norm can help to upper bound both terms and eventually ensure the TD-stability. Considering that layer norm is also widely present in the CIR architecture, the natural question is whether addition of Tanh is really crucial for stability or it mostly further complements the layer-norm regularization. Moreover, looking at Figure 5 (right), we can see that replacing Tanh activation with Layer Norm doesn’t make a significant difference (means are quite close to each other, and confidence intervals overlap almost entirely).

Moreover, the usage of U-net architecture was not properly ablated. In particular, it is not clear if by taking Bro architecture but with Tanh for the first layer activation function and convex Q-learning we can achieve similar results on HumanoidBench. The authors motivate the usage of U-net, because of its capability to perform "multi-scale feature extraction", at the same time you do not perform downsampling and upsampling, which I believe contradicts the aforementioned motivation. Have you tried a U-net variant with actual downsampling and upsampling?

Finally, I am not fully convinced by a current proof of convex-Q learning convergence. The proof is only shown for one $Q^A$ function by considering that the action in the target value is $a^* = \arg \max Q^A_t(s,a)$, which indeed allows to obtain that $\mathbb{E}[F_t^Q | P_t] \leq \gamma \|\Delta_t\|$ for $F_t^Q$ defined with $Q^A$, however this argument will not work if $F_t^Q$ is redefined with $Q_t^B$. The statement of the theorem still looks reasonable, so a few corrections and clarifications might suffice to make it sound.

Minor:

- Can you use bold in every table to have a clear picture on which tasks CIR improves and on which it is less performant
- L108: I don’t see how tanh is connected to sample-efficiency, at least the link is not explicit
- I would also consider longer training time for many benchmarks to make sure that the final results are stable. For instance, for many DMC hard problems and HumanoidBench the curves continue to improve for many baselines and CIR.
- L874: not sure that you can extract $(r+ \gamma\phi(s’)^T \theta  - \phi(s)^T \theta)$ out of expectation
- L901: it can be even an easier expression by taking directly $Var[X] \leq \mathbb{E} X^2$
- Table 10: sometimes pure tanh significantly degrades the performance, which should be further studied and discussed
- L1033: TYPO should be $\zeta_t(x) = 0$
- L1046-1047: "The condition (4) of the lemma holds by the condition (6) in our theorem." As $F_t$ is not yet defined at this point, it should be shown more explicitly.

**Questions:**

- In Eq.10, how is $a’$ chosen?
- Have you considered crossQ as a baseline? They use batch normalisation, which allows them to avoid target computations.
- For figure 5 you use computation time as a criterion,  in this case can you please specify what hardware and software (PyTorch or JAX?) do you use for all baselines?
- Why layer norm can be seen as a parameter regulariser like in Theorem 4.5?
- Are you performing 10 seeds for all ablation studies?
- Acrobot-swingup looks like an anomaly case for CIR, can you give more insights on what is going on with this environment?

---

### Official Review · Reviewer_XQKV · 2025-10-23

**Soundness:** 2
**Presentation:** 2
**Contribution:** 2
**Rating:** 2
**Confidence:** 4

**Summary:**

The paper proposes a novel architecture for sample efficient off-policy learning. The proposed method incorporates changes to the input representation via a tanh block, a u-net-like critic architecture, convex Q-learning instead of clipped double Q learning, and SRM instead of an increased UTD ratio. Of the design choices, strong focus is put on analyzing the impact of the tanh function in the theoretical contributions.

**Strengths:**

The paper presents thorough experiments on their proposed method.

**Weaknesses:**

Theoretical results on representation learning within a fixed representation linear function approximation setting seem unsuited to shine light on the effectiveness of the method. In addition, the analysis assumes that the tanh function is applied after the representation function $\phi$. However, most of the representation learning in the empirical method happens *after* the tanh function.
The method analyzed here is more akin to OFN (Hussing et al.) or hyperspherical normalization (Lee et al.).

While the focus of the writing is on the tanh activation, there are several additional changes introduced. The tanh change does not seem to be the one with the highest performance impact, yet it is the only one theoretically justified. In addition, the comparison against Simba and BRO has limited merit given the framing of the paper: if the tanh is the important contribution, it would make much more sense to evaluate the Simba and BRO architectures with and without the tanh function.

There is a growing trend to simply propose new architectures in the literature (the aforementioned papers being prime examples) but I think there is limited value in comparing bags of design decisions against one another instead of making controlled and carefully ablated additions to existing architectures. Making choices such as SRM vs UTD further muddies the comparison.

While the paper does compare against strong method published over the last year, it does not compare against the newest version of Simba (ICML 2025) or the Mr.Q architecture. Both outperform baselines reported here and should make for a more up-to-date point of comparison. Both papers provide results, so this does not require additional experimentation.

Simba-v2 results: https://github.com/dojeon-ai/SimbaV2/tree/master/results

Mr.Q results: https://github.com/facebookresearch/MRQ

OFN: Dissecting Deep RL with High Update Ratios: Combatting Value Divergence, RLC 2024, Hussing et al.

Hyperspherical normalization: Hyperspherical Normalization for Scalable Deep Reinforcement Learning, ICML 2025, Lee et al.

**Questions:**

Should the tanh function be applied to the output of the neural network representation learning, given the theoretical results?

How does tanh normalization compare to other forms of input normalization, such as mean/std and min max normalization.

---

### Official Review · Reviewer_WeuV · 2025-10-30

**Soundness:** 1
**Presentation:** 3
**Contribution:** 1
**Rating:** 2
**Confidence:** 5

**Summary:**

This work studies representations in deep reinforcement learning, specifically representations early into the network architecture. The paper presents a new architectural module that is placed on top of the input called CIR where the key components is a tanh activation function. A theoretical section provides various results on tanh activations in linear RL. Then, the provides an experimental section on four different benchmarks comparing against state-of-the-art baselines.

**Strengths:**

**Motivation**
* The problem of studying architectural components that improve deep reinforcement learning is important and well motivated.

**Clarity**
* The text is well written and easy to follow.

**Related Work**
* The related work section is quite extensive.

**Experimental Design and Analyses**
* The choices of benchmarks and baselines are appropriate and extensive.

**Weaknesses:**

**Motivation**
* There are two key issues that I see with the motivation of the method specifically.
    * It is unclear to me why previous studies on regularization would not immediately extend to early neural network layers and why specifically constraining the inputs is important. This is not clearly communicated in the text either.
    * The choice of tanh activations rather than any other function seems arbitrary and not well motivated. The properties that the tanh function brings are satisfied by simpler functions as well. Tanh seems to be a non-linear way to regularize but as I will outline the evidence for the need of this is not sufficiently strong.

**Clarity**
* The experimental results are presented in plots that are a bit difficult to read. For example, I don’t see any shaded regions in Figure 4. If the standard errors are small, I think the paper would benefit from simply removing the weak baselines since they don’t tell us much about the performance of the presented algorithm.
* I think the number of seeds should be reported in the main text, not the Appendix.
* I find the term scaling confusing in section 6.3. There is a difference between making a network large enough so it can sufficiently represent the required function and vastly overparameterizing it. When other domains talk about scaling, they refer to moving from millions of parameters to billions. That is clearly not the case here.

**Related Work**
* There are a few minor nitpicks that I have with the treatment of related work
    * The paper repeatedly states that using larger networks is a challenge and cites Bjorck et al. [1], and then states that the first paper to attempt and resolve this is Nauman et al. [2]. This is a misrepresentation as Bjorck et al is quite literally about resolving this issue.

**Novelty**
* In general, I think the novelty of the approach is limited as the effects of various components that are being applied have already been studied extensively. These are continually referred to as novel components but they already exist in the literature.
    * The effects of layer normalization are well studied [2, 3, 8]
    * So are the effects of RMSnorms and related ones as presented in section 5.1[4, 5, 6].
    * Specifically [5, 8] study normalization as a way to mitigate the divergence effects from high dimensional spaces. It might make sense to include them and to distinguish the suggested normalization approach if possible.
    * Skip connections have previously been introduced as well. [2]
    * Convex combinations of Q-values have also been studied before [7].
* Other parts of the module that are being proposed are often already considered in various implementations and I would not claim they are novel. While there is value in analyzing tricks that people have coded into their implementations, yet such analysis should be extensive. That is not the case for section 5.3 for instance.

This would be fine if the performance gains were very large but as I will outline later I have my doubts about the effectiveness of this rather complex approach.

**Theoretical Results**
* The theoretical analysis is not done on top of the early layers of some complex functions but instead on the output of the feature encoder in which the function is linear. In deep RL, that would correspond to the second to last layers not the initial one. This is disconnect from what the paper attempts to show.
* Various of the theoretical results have strong assumptions.
    * These assumptions may allow for much more general statements than the ones made in the manuscript. For example, with the assumption on the weights in Theorem 4.2, any transformation that does not change the direction of the basis functions would suffice.
    * They also make some of the results very weak. The proof of Theorem 4.2 seems to rely on c being very small such that the weights are close to 0. Thus, the whole matrix over which we take the tanh is 0. That seems not very useful in practice. It is also not clear to me why this would be true at any point during the fitting of our Q-function.
* The assumptions make the theoretical results quite weak. For example, in Theorem 4.2 even if I initialize my weights with an identity matrix, and $W \phi$ is a selection of uncorrelated vectors, a single gradient step might correlate them.
* In section 4, when talking about Theorem 4.4 and Theorem 4.5, the relation to the true Q-function is unclear. The text states that it is unclear whether the TD update converges to a global minimum. It is not clear to me which global minimum is being talked about as it is not defined in the text. The goal should be to learn an optimal Q-function. It is unclear to me that Theorem 4.4 would approximate that. There is a difference between a globally optimal solution to the equation of interest and the global minimizer in Theorem 4.5. The regularization induces convexity which allows us to minimize the proposed objective. I don’t see how there is a guarantee that that minimizer approaches the minimizer that we are looking for, namely the one of the optimal Q-function. As an example, one can always set lambda to be very large, the given objective is convex and thus, one can find a global minimum. That minimum however might be all weights equal 0 which is clearly not always close to the true Q-function weights. I only skimmed the proof but the “global minimum” that the proof seems to use is the one of the regularized objective which is not sufficient unless we make more assumptions.

**Method design**
* Several key components seem rather arbitrary or incorrectly motivated. Some of the statements made to motivate these components are likely incorrect or not well enough explained and supported with evidence.
    * L 221: “latter layers can be naturally regularized” if the initial representations are constrained, since the magnitude of the input representations are controlled” - The magnitude of the input being constrained does not affect later layers. In general, observational inputs are limited by physical constraints in many of the suggested benchmarks, yet we see the need for regularization.
    * L 223: “it may be challenging for the gradient to backpropagate and the gradient may vanish before the initial layers can be well-updated” - the paper argues that one benefit of the approach is to limit gradient variance but then does not seem to use that fact.
    * L243: “based on Theorem 4.5, additional regularization terms are needed to acquire global convergence. Since regularizing the parameters is inherently equivalent to regularizing the representations, we further regularize representations for better stability by involving layer normalization, which is proven effective in prior works” - Layer normalization does not create a convex minimizer and it is unclear why it would be a substitute for L2 regularization. These two have inherently different effects.
    * L307: “meanwhile, constraining initial representations with Tanh can be conservative since it constrains the initial representations” - It is unclear to me why this would lead to pessimism on the predictions since the initial representations are followed by a rather complex function object. Limiting the observation space does not induce pessimism.

**Experimental Design and Analyses**
* It is unclear to me why we wouldn’t compare against other methods that use normalization on ODRL. These are the approaches that are most similar.

**Empirical Claims and Evidence**
* The paper makes various claims about experiments that are not sufficiently supported.
    * The paper claims that the introduced module provides higher sample efficiency than other methods (L 085), however no experiment for this is presented.
    * The paper claims that performance on DMC easy exceeds all other baselines but it seems that results are within variance of BRO fast and TD-MPC2.
    * In section 6.3 the paper argues that SimBa underperforms CIR. However, 4/6 results are within variance. I think it would be better practice to highlight statistically significant differences rather than bolding the mean.
* A large chunk of the paper argues the need for a tanh activation however, simply using LayerNorm as suggested by other works seems to be just as effective. Thus, it is not clear how the effects of tanh and the experiments don’t make this clear.
* Similarly, the skip connections don’t seem to bring any benefits in the ablation so it is not clear why they were included. It seems they might just increase the complexity of the method. If we removed both the skip connections and the Tanh, they method more or less reduces to existing layer norm approaches [2]
* Overall, the experimental results seem comparable to previous approaches which is likely because the method looks somewhat similar to what prior work does.

[1] Towards Deeper Deep Reinforcement Learning. Bjorck et al. 2021.
[2] Overestimation, Overfitting, and Plasticity in Actor-Critic: the Bitter Lesson of Reinforcement Learning. Nauman et al. 2024.
[3] Disentangling the Causes of Plasticity Loss in Neural Networks. Lyle et al. 2024.
[4] Is High Variance Unavoidable in RL? A Case Study in Continuous Control. Bjorck et al. 2022.
[5] Dissecting Deep RL with High Update Ratios: Combatting Value Divergence. Hussing et al. 2024.
[6] SimBa: Simplicity Bias for Scaling Up Parameters in Deep Reinforcement Learning. Lee et al. 2025.
[7] Off-Policy Deep Reinforcement Learning without Exploration. Fujimoto et al. 2019.
[8] Understanding, Predicting and Better Resolving Q-Value Divergence in Offline-RL. Yue et al. 2023

**Questions:**

Q1: It is unclear to me why Convex Q-learning is called that. I don’t see how taking non-linear operations over a highly complex function would be convex. Is this because it is a convex combination of 2 Q-values? If so, I would recommend being more careful with the naming because the Q-values are not convex and thus, neither is the Q-learning.

---

### Official Review · Reviewer_3fSP · 2025-10-31

**Soundness:** 3
**Presentation:** 2
**Contribution:** 1
**Rating:** 2
**Confidence:** 5

**Summary:**

This paper proposes an input-preprocessing technique (CIR) alongside a UNet-style skip connection and modified Q-learning update to improve the performance of deep RL agents in continuous control benchmarks. A variety of ablations are run to demonstrate the utility of these additions.

**Strengths:**

The results are presented clearly and the experiments conducted are sound and varied.

**Weaknesses:**

I find that the main weakness of the paper is the limited novelty.
Many of the ideas in CIR have been considered before (or are a small extension of existing ideas) and some are not motivated clearly. The paper primarily combines these techniques and there seems to be a lack of focus.
In more detail:
- The convex Q-learning update is a small extension to using the mean Q-value from two critics, which is prominently used in BRO for example. It's unclear how much this extension adds compared to using the mean. Also, in the appendix, I noticed that a different convex interpolation factor is needed for DMC vs. HumanoidBench, which suggests the method is not robust to different environments.

- UNet-style skip connections are used. The benefits of this method seem to be limited judging by the ablation done in Fig.5 (right). I am also unsure of the specific motivation for this architectural choice in a deep RL setting because I have usually seen it used in the context of image-reconstruction-style networks where the UNet structure intuitively makes sense to match inputs and outputs.

- Sample multiple reuse (SMR) is used directly as-is from its original paper. This seems like a lesser known technique in the RL commmunity though.

- Using tanh on the inputs may be an exception to this criticism. I have not seen this technique used before. Unfortunately, I am also doubtful on the specific utility of tanh vs. LayerNorm. In Fig.5 (right), LayerNorm on the inputs seems to perform almost just as well as tanh.


Concerning the experiments: The SimBaV2 baseline is notably absent from the evaluations, a stronger baseline than those currently in the paper on these continuous control benchmarks.

I have outlined a few suggestions in the "Questions" section to improve the paper.

**Questions:**

Clarifications:
- When tanh is applied to the inputs, is observation normalization using running mean and variance statistics also done? If so, is observation normalization done before or after tanh?

Suggestions (not directly impacting the score):

I would suggest focusing the paper on either SMR or the tanh on inputs aspects. Those topics seems less studied in RL and perhaps some further investigation into these areas could lead to the development of novel and effective algorithms.

It seems like the authors intended to do this with the CIR method: In the introduction of the paper, the main contribution is framed as the CIR method but later in the paper many other techniques are introduced (SMR, skip connection, convex Q-learning) which seem completely unrelated.
I would encourage the authors to focus on this direction and removing these unrelated additional techniques. Doing more in-depth studies of the tanh input preprocessing would be welcome. For example, checking how network predictions/optimization is impacted when feeding in OOD inputs with/without tanh preprocessing. This was suggested in the introduction as a potential advantage of tanh but it wasn't specifically demonstrated.
Some other ideas that come to mind also include revisiting tile coding or RBF features, which may have other advantages due to sparsity or locality.

---

### Note · Authors · 2026-01-04

I have read and agree with the venue's withdrawal policy on behalf of myself and my co-authors.